# ON THE ROLE OF PLANNING IN MODEL-BASED DEEP REINFORCEMENT LEARNING

**Jessica B. Hamrick,**[*] **Abram L. Friesen, Feryal Behbahani, Arthur Guez, Fabio Viola, Sims Witherspoon, Thomas Anthony, Lars Buesing, Petar Veličković, Théophane Weber**[*]
DeepMind, London, UK

## ABSTRACT

Model-based planning is often thought to be necessary for deep, careful reasoning and generalization in artificial agents. While recent successes of model-based reinforcement learning (MBRL) with deep function approximation have strengthened this hypothesis, the resulting diversity of model-based methods has also made it difficult to track which components drive success and why. In this paper, we seek to disentangle the contributions of recent methods by focusing on three questions: (1) How does planning benefit MBRL agents? (2) Within planning, what choices drive performance? (3) To what extent does planning improve generalization? To answer these questions, we study the performance of MuZero [58], a state-of-the-art MBRL algorithm with strong connections and overlapping components with many other MBRL algorithms. We perform a number of interventions and ablations of MuZero across a wide range of environments, including control tasks, Atari, and 9x9 Go. Our results suggest the following: (1) Planning is most useful in the learning process, both for policy updates and for providing a more useful data distribution. (2) Using shallow trees with simple Monte-Carlo rollouts is as performant as more complex methods, except in the most difficult reasoning tasks. (3) Planning alone is insufficient to drive strong generalization. These results indicate where and how to utilize planning in reinforcement learning settings, and highlight a number of open questions for future MBRL research.

Model-based reinforcement learning (MBRL) has seen much interest in recent years, with advances yielding impressive gains over model-free methods in data efficiency [12, 15, 25, 76], zero- and few-shot learning [16, 37, 60], and strategic thinking [3, 62, 63, 64, 58]. These methods combine planning and learning in a variety of ways, with *planning* specifically referring to the process of using a learned or given model of the world to construct imagined future trajectories or plans.

Some have suggested that models will play a key role in generally intelligent artificial agents [14, 50, 55, 56, 57, 67], with such arguments often appealing to model-based aspects of human cognition as proof of their importance [24, 26, 28, 41]. While the recent successes of MBRL methods lend evidence to this hypothesis, there is huge variance in the algorithmic choices made to support such advances. For example, planning can be used to select actions at evaluation time [e.g., 12] and/or for policy learning [e.g., 34]; models can be used within discrete search [e.g., 58] or gradient-based planning [e.g., 25, 29]; and models can be given [e.g., 45] or learned [e.g., 12]. Worryingly, some works even come to contradictory conclusions, such as that long rollouts can hurt performance due to compounding model errors in some settings [e.g., 34], while performance continues to increase with search depth in others [58]. Given the inconsistencies and non-overlapping choices across the literature, it can be hard to get a clear picture of the full MBRL space. This in turn makes it difficult for practitioners to decide which form of MBRL is best for a given problem (if any).

The aim of this paper is to assess the strengths and weaknesses of recent advances in MBRL to help clarify the state of the field. We systematically study the role of planning and its algorithmic design choices in a recent state-of-the-art MBRL algorithm, MuZero [58]. Beyond its strong performance, MuZero's use of multiple canonical MBRL components (e.g., search-based planning, a learned model, value estimation, and policy optimization) make it a good candidate for building intuition about the roles of these components and other methods that use them. Moreover, as discussed in the

---

[*]Correspondence addressed to: {`jhamrick,theophane`}@`google.com`

next section, MuZero has direct connections with many other MBRL methods, including Dyna [67], MPC [11], and policy iteration [33].

To study the role of planning, we evaluate overall reward obtained by MuZero across a wide range standard MBRL environments: the DeepMind Control Suite [70], Atari [8], Sokoban [51], Minipacman [22], and 9x9 Go [42]. Across these environments, we consider three questions. (1) For what purposes is planning most useful? Our results show that planning—which can be used separately for policy improvement, generating the distribution of experience to learn from, and acting at test-time—is most useful in the learning process for computing learning targets and generating data. (2) What design choices in the search procedure contribute most to the learning process? We show that deep, precise planning is often unnecessary to achieve high reward in many domains, with two-step planning exhibiting surprisingly strong performance even in Go. (3) Does planning assist in generalization across variations of the environment—a common motivation for model-based reasoning? We find that while planning can help make up for small amounts of distribution shift given a good enough model, it is not capable of inducing strong zero-shot generalization on its own.

## 1 BACKGROUND AND RELATED WORK

Model-based reinforcement learning (MBRL) [9, 26, 47, 49, 74] involves both learning and planning. For our purposes, *learning* refers to deep learning of a model, policy, and/or value function. *Planning* refers to using a learned or given model to construct trajectories or plans. In most MBRL agents, learning and planning interact in complex ways, with better learning usually resulting in better planning, and better planning resulting in better learning. Here, we are interested in understanding how differences in planning affect both the learning process and test-time behavior.

MBRL methods can be broadly classified into *decision-time* planning, which use the model to select actions, and *background* planning, which use the model to update a policy [68]. For example, model-predictive control (MPC) [11] is a classic decision-time planning method that uses the model to optimize a sequence of actions starting from the current environment state. Decision-time planning methods often feature robustness to uncertainty and fast adaptation to new scenarios [e.g., 76], though may be insufficient in settings which require long-term reasoning such as in sparse reward tasks or strategic games like Go. Conversely, Dyna [67] is a classic background planning method which uses the the model to simulate data on which to train a policy via standard model-free methods like Q-learning or policy gradient. Background planning methods often feature improved data efficiency over model-free methods [e.g., 34], but exhibit the same drawbacks as model-free approaches such as brittleness to out-of-distribution experience at test time.

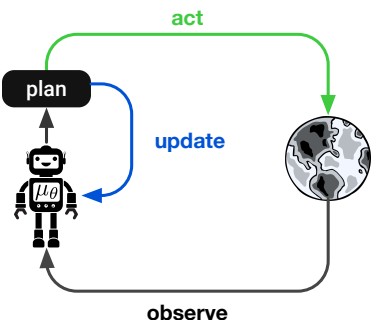

Figure 1: Model-based approximate policy iteration. The agent **updates** its policy using targets computed via planning and optionally **acts** via planning during training, at test time, or both.

A number of works have adopted hybrid approaches combining both decision-time and background planning. For example, Guo et al. [23], Mordatch et al. [48] distill the results of a decision-time planner into a policy. Silver et al. [61], Tesauro & Galperin [71] do the opposite, allowing a policy to guide the behavior of a decision-time planner. Other works do both, incorporating the distillation or imitation step into the learning loop by allowing the distilled policy from the previous iteration to guide planning on the next iteration, illustrated by the "update" arrow in Figure 1. This results in a form of approximate policy iteration which can be implemented both using single-step [40, 54] or multi-step [17, 18] updates, the latter of which is also referred to as expert iteration [3] or dual policy iteration [66]. Such algorithms have succeeded in board games [3, 4, 63, 64], discrete-action MDPs [27, 51, 58] and continuous control [43, 45, 65].

In this paper, we focus our investigation on MuZero [58], a state-of-the-art member of the approximate policy iteration family. MuZero is a useful testbed for our analysis not just because of its strong performance, but also because it exhibits important connections to many other works in the MBRL literature. For example, MuZero implements a form of approximate policy iteration [18] and

has close ties to MPC in that it uses MPC as a subroutine for acting. There is also an interesting connection to recent Dyna-style MBRL algorithms via regularized policy optimization: specifically, these methods simulate data from the model and then update the policy on this data using TRPO [34, 39, 46, 52]. Recent work by Grill et al. [20] showed that the MuZero policy update approximates TRPO, making the learning algorithm implemented by these Dyna-style algorithms quite similar to that implemented by MuZero. Finally, the use of value gradients [10, 19, 25, 29] leverages model gradients to compute a policy gradient estimate; this estimate is similar (without regularization) to the estimate that MCTS converges to when used as a policy iteration operator [20, 75]. Thus, our results with MuZero have implications not just for its immediate family but to MBRL methods more broadly.

In contrast to other work in MBRL, which focuses primarily on data efficiency [e.g., 34, 36] or model learning [e.g., 12], our primary concern in this paper is in characterizing the role of planning with respect to reward after a large but fixed number of learning steps, as well as to zero-shot generalization performance (however, we have also found our results to hold when performing the same experiments and measuring approximate regret). We note that MuZero can also be used in a more data-efficient manner ("MuZero Reanalyze") by using the search to re-compute targets on the same data multiple times [58], but leave exploration of its behavior in this regime to future work.

Our analysis joins a number of other recent works that seek to better understand the landscape of MBRL methods and the implications of their design choices. For example, Chua et al. [12] perform a careful analysis of methods for uncertainty quantification in learned models. Other research has investigated the effect of deep versus shallow planning [32, 35], the utility of parametric models in Dyna over replay [73], and benchmark performance of a large number of popular MBRL algorithms in continuous control tasks [74]. Our work is complementary to these prior works and focuses instead on the different ways that planning may be used both during training and at evaluation.

## 2  PRELIMINARIES: OVERVIEW OF MUZERO

MuZero uses a learned policy and learned value, transition and reward models within Monte-Carlo tree search (MCTS) [13, 38] both to select actions and to generate targets for policy learning (Figure 1). We provide a brief overview of MuZero here and refer readers to Appendix A and Schrittwieser et al. [58] for further details. Algorithm 1 and 2 present pseudocode for MuZero and MCTS, respectively.

**Model**  MuZero plans in a hidden state space using a learned model $\mu_\theta$ parameterized by $\theta$ and comprised of three functions. At timestep $t$, the *encoder* embeds past observations into a hidden state, $s_t^0 = h_\theta(o_1, \ldots, o_t)$. Given a hidden state and an action in the original action space, the (deterministic) recurrent *dynamics* function predicts rewards and next states, $r_{\theta,t}^k, s_t^k = g_\theta(s_t^{k-1}, a_t^{k-1})$, where $k$ is the number of imagined steps into the future starting from a real observation at time $t$. In addition, the *prior* (not used in the classic Bayesian sense of the term) predicts a policy and value for a given hidden state, $\pi_{\theta,t}^k, v_{\theta,t}^k = f_\theta(s_t^k)$, and is used to guide the tree search.

**Search**  Beginning at the root node $s_t^0$, each simulation traverses the search tree according to a *search policy* until a previously unexplored action is reached. The search policy is a variation on the pUCT rule [53, 38] that balances exploitation and exploration, and incorporates the prior to guide this (see Equation 1 in Section A.3). After selecting an unexplored action $a^\ell$ at state $s_t^\ell$, the tree is expanded by adding a new node with reward $r_{\theta,t}^{\ell+1}$, state $s_t^{\ell+1}$, policy $\pi_{\theta,t}^{\ell+1}$, and value $v_{\theta,t}^{\ell+1}$ predicted by the model. The value and reward are used to form a bootstrapped estimate of the cumulative discounted reward, which is backed up to the root, updating the estimated return $Q$ and visit count $N$ of each node on the path. After $B$ simulations, MCTS returns a value $v_t^{\text{MCTS}}$ (the average cumulative discounted reward at the root) and policy $\pi_t^{\text{MCTS}}$ (a function of the normalized count of the actions taken at the root during search).

**Acting**  After search, an action is sampled from the MCTS policy, $a_t \sim \pi_t^{\text{MCTS}}$, and is executed in the environment to obtain reward $r_t^{\text{env}}$. Data from the search and environment are then added to a replay buffer for use in learning: $\{o_t, a_t, r_t^{\text{env}}, \pi_t^{\text{MCTS}}, v_t^{\text{MCTS}}\}$.

**Learning**  The model is jointly trained to predict the reward, policy, and value for each future timestep $k = 0 \ldots K$. The reward target is the observed environment reward, $r_{t+k}^{\text{env}}$. The policy target

is the MCTS-constructed policy $\pi_{t+k}^{\mathrm{MCTS}}$. The value target is the $n$-step bootstrapped discounted return $z_t = r_{t+1}^{\mathrm{env}} + \gamma r_{t+2}^{\mathrm{env}} + \cdots + \gamma^{n-1} r_{t+n}^{\mathrm{env}} + \gamma^n v_{t+n}^{\mathrm{MCTS}}$. For reward, value, and policy losses $\ell^r, \ell^v$, and $\ell^p$, respectively, the overall loss is then $\ell_t(\theta) = \sum_{k=0}^{K} \ell^r(r_{\theta,t}^k, r_{t+k}^{\mathrm{env}}) + \ell^v(v_{\theta,t}^k, z_{t+k}) + \ell^p(\pi_{\theta,t}^k, \pi_{t+k}^{\mathrm{MCTS}})$. Note that MuZero is not trained to predict future observations or hidden states: the learning signal for the dynamics comes solely from predicting future rewards, values, and policies. We do, however, perform additional experiments in Section D.2 in which we train MuZero to also predict observations via an additional reconstruction loss; our results indicate that the observation-based model is slightly more accurate but does not qualitatively change MuZero's behavior.

## 3 HYPOTHESES AND EXPERIMENTAL METHODS

Our investigation focuses on three key questions: (1) How does planning drive performance in MuZero? (2) How do different design choices in the planner affect performance? (3) To what extent does planning support generalization? To answer these questions, we manipulated a number of different variables across our experiments (Figure 2, see also Section B.1): the maximum depth we search within the tree ($D_{\mathrm{tree}}$), the maximum depth we optimize the exploration-exploitation tradeoff via pUCT ($D_{\mathrm{UCT}}$), the search budget ($B$), the model (learned or environment simulator), and the planning algorithm itself (MCTS or breadth-first search).

**(1) Overall contributions of planning** Performance in many model-free RL algorithms is driven by computing useful policy improvement targets. We hypothesized that, similarly, using the search for policy improvement (as opposed to exploration or acting) is a primary driver of MuZero's performance, with the ability to compare the outcome of different actions via search enabling even finer-grained and therefore more powerful credit assignment. To test this hypothesis, we implemented several variants of MuZero which use search either for learning, for acting, or both. When learning, we compute $\pi^{\mathrm{MCTS}}$ either using full tree search ($D_{\mathrm{tree}} = \infty$) or using only one step of lookahead ($D_{\mathrm{tree}} = 1$). When acting, we sample actions either from the policy prior $\pi_{\theta,t}$ or from $\pi^{\mathrm{MCTS}}$ computed using $D_{\mathrm{tree}} = \infty$. These choices allow us to define the following variants (see table in Figure 3): "One-Step", which trains and uses the model in a one-step way to generate learning targets, similar to having a Q-function; "Learn", which uses the search only for learning; "Data", which uses the search only to select actions during training while using the model in a one-step way to generate targets; "Learn+Data", which uses the search both for learning and for action selection during training; and "Learn+Data+Eval", which corresponds to full MuZero. See Section B.2 for more details about these variants.

**(2) Planning for learning** One feature of MCTS is its ability to perform "precise and sophisticated lookahead" [58]. To what extent does this lookahead support learning stronger policies? We hypothesized that more complex planning like tree search—as opposed to simpler planning, like random shooting—and deeper search is most helpful for learning in games like Go and Sokoban, but less helpful for the other environments. To test this, we manipulated the tree depth ($D_{\mathrm{tree}}$), UCT depth ($D_{\mathrm{UCT}}$), and search budget[1] ($B$). Note that our aim with varying $D_{\mathrm{UCT}}$ is to evaluate the effect of *simpler* versus more complex planning, rather than the effect of exploration.

**(3) Generalization in planning** Model-based reasoning is often invoked as a way to support generalization and flexible reasoning [e.g., 26, 41]. We similarly hypothesized that given a good model, planning can help improve zero-shot generalization. First, we evaluated the ability of the individual model components to generalize to new usage patterns. Specifically, we evaluated pre-trained agents using: larger search budgets than seen during

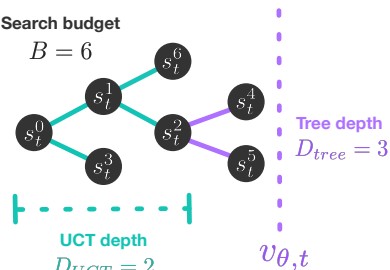

Figure 2: For nodes at depth $d < D_{\mathrm{UCT}}$, we select actions according to pUCT (Section 2), while for nodes at depth $D_{\mathrm{UCT}} \leq d < D_{\mathrm{tree}}$, we select actions by sampling from $\pi_{\theta,t}$. Nodes at depth $d = D_{\mathrm{tree}}$ (and deeper) are not expanded; instead, we stop the search and backup using $v_{\theta,t}$. The search budget $B$ is equal to the number of nodes in the tree aside from the root $s_t^0$.

---

[1]MuZero's policy targets suffer from degeneracies at low visit counts [20, 27]; to account for this, we used an MPO-style update [1] in the search budget experiments, similar to Grill et al. [20]. See Section B.3.

training, either the learned model or the environment simulator, and either MCTS or breadth-first search (see Section B.4). Second, we tested generalization to unseen scenarios by evaluating pre-trained Minipacman agents of varying quality (assessed by the number of unique mazes seen during training) on novel mazes drawn from the same or different distributions as in training.

# 4  RESULTS

We evaluated MuZero on eight tasks across five domains, selected to include popular MBRL environments with a wide range of characteristics including episode length, reward sparsity, and variation of initial conditions. First, we included two Atari games [8] which are commonly thought to require long-term coordinated behavior: **Ms. Pacman** and **Hero**. We additionally included **Minipacman** [51], a toy version of Ms. Pacman which supports procedural generation of mazes. We also included two strategic games that are thought to heavily rely on planning: **Sokoban** [22, 51] and **9x9 Go** [42]. Finally, because much work in MBRL focuses on continuous control [e.g., 74], we also included three tasks from the DeepMind Control Suite [70]: **Acrobot** (Sparse Swingup), **Cheetah** (Run), and **Humanoid** (Stand). We discretized the action space of the control tasks as in Tang & Agrawal [69], Grill et al. [20]. Three of these environments also exhibit some amount of stochasticity and partial observability: the movement of ghosts in Minipacman is stochastic; Go is a two-player game and thus stochastic from the point of view of each player independently; and using a limited number of observation frames in Atari makes it partially observable (e.g., it is not possible to predict when the ghosts in Ms. Pacman will change from edible to dangerous). Further details of all environments are available in Appendix C.

## 4.1  BASELINES

Before beginning our analysis, we tuned MuZero for each domain and ran baseline experiments with a search budget of $B = 10$ simulations in Minipacman, $B = 25$ in Sokoban, $B = 150$ in 9x9 Go, and $B = 50$ in all other environments. Additional hyperparameters for each environment are available in Appendix C and learning curves in Section D.3; unless otherwise specified, all further experiments used the same hyperparameters as the baselines. We obtained the following median final scores, computed using the last 10% of steps during training (median across ten seeds): 620.07 on Acrobot, 885.57 on Cheetah, 787.98 on Humanoid, 29916.97 on Hero, 45346.71 on Ms. Pacman, 310.1 on Minipacman, 0.97 on Sokoban (proportion solved), and 0.72 on 9x9 Go (proportion games won against Pachi 10k [7], a bot with strong amateur play). These baselines are all very strong, with the Atari and Control Suite results being competitive with state-of-the-art [22, 31, 58].

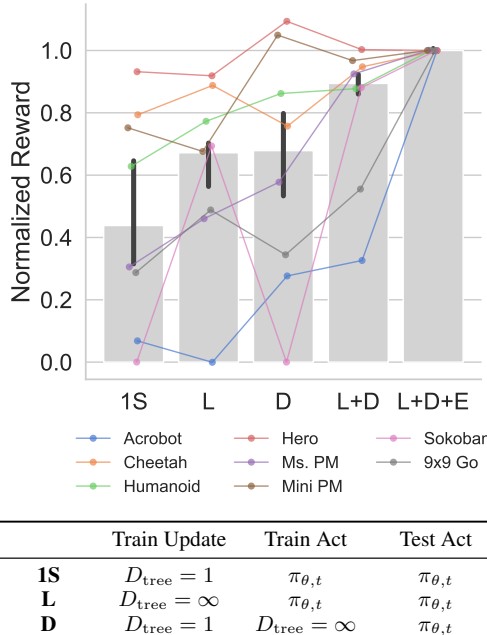

| | Train Update | Train Act | Test Act |
|---|---|---|---|
| **1S** | $D_{\text{tree}} = 1$ | $\pi_{\theta,t}$ | $\pi_{\theta,t}$ |
| **L** | $D_{\text{tree}} = \infty$ | $\pi_{\theta,t}$ | $\pi_{\theta,t}$ |
| **D** | $D_{\text{tree}} = 1$ | $D_{\text{tree}} = \infty$ | $\pi_{\theta,t}$ |
| **L+D** | $D_{\text{tree}} = \infty$ | $D_{\text{tree}} = \infty$ | $\pi_{\theta,t}$ |
| **L+D+E** | $D_{\text{tree}} = \infty$ | $D_{\text{tree}} = \infty$ | $D_{\text{tree}} = \infty$ |

Figure 3: Contributions of planning to performance, where 0.0 is the performance attained by a randomly initialized policy (Table 8), and 1.0 that obtained by full MuZero (Table 7). Grey bars show medians across environments, and error bars show 95% confidence intervals of the median. 1S=One-Step, L=Learn, D=Data, E=Eval. See Figure 10 for environment error bars.

## 4.2  OVERALL CONTRIBUTIONS OF PLANNING

We first compared vanilla MuZero to different variants which use search in varying ways, as described in Section 3. To facilitate comparisons across environments, we normalized scores to lie between the performance attained by a randomly initialized policy (Table 8) and the full version of MuZero (Table 7). Figure 3 shows the results. Across environments, the "One-Step" variant has a median strength of 46.7% ($N = 80$).

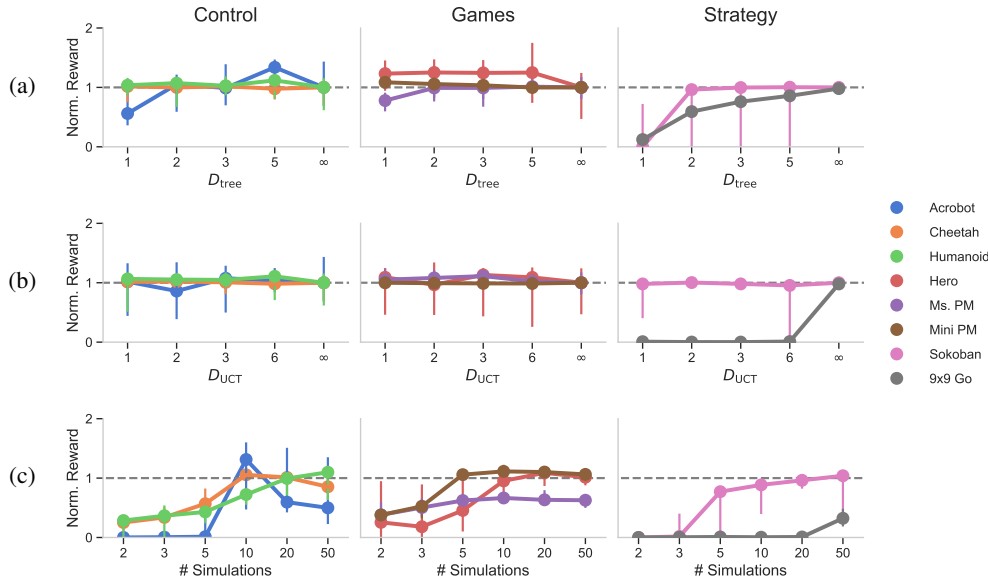

Figure 4: Effect of design choices on the strength of the policy prior. All colored lines show median normalized reward across ten seeds (except Go, which uses five seeds), with error bars indicating min and max seeds. Rewards are normalized by the median scores in Table 9. All agents use search for learning and acting during training only. (a) Reward as a function of $D_{\text{tree}}$. Here, $D_{\text{UCT}} = D_{\text{tree}}$ and the number of simulations is the same as the baseline. (b) Reward as a function of $D_{\text{UCT}}$. Here, $D_{\text{tree}} = \infty$ and the number of simulations is the same as the baseline. (c) Reward as a function of search budget during learning. Here, $D_{\text{UCT}} = 1$ and $D_{\text{tree}} = \infty$ (except in Go, where $D_{\text{UCT}} = \infty$).

Although this variant is not entirely model-free, it does remove much of the dependence on the model, thus establishing a useful minimal-planning baseline to compare against. Using a deeper search solely to compute policy updates ("Learn") improves performance to 68.5% ($N = 80$). The "Data" variant—where search is only used to select actions—similarly improves over "One-Step" to 66.7% ($N = 80$). These results indicate both the utility in training a multi-step model, and that search may also drive performance by enabling the agent to learn from a different state distribution resulting from better actions (echoing other recent work leveraging planning for exploration [45, 60]). Allowing the agent to both learn and act via search during training ("Learn+Data") further improves performance to a median strength of 90.3% ($N = 80$). Finally, search at evaluation ("Learn+Data+Eval") brings the total up to 100%.

Using the search for learning, for acting during training, and for acting at test time appears to provide complementary benefits (see also Table 12). This confirms our hypothesis that an important benefit of search is for policy learning, but also highlights that search can have positive impacts in other ways, too. Indeed, while some environments benefit more from a model-based learning signal (Cheetah, Sokoban, Go) others benefit more from model-based data (Hero, Minipacman, Humanoid, Ms. Pacman, Acrobot). Combining these two uses enables MuZero to perform well across most environments (though Hero and Minipacman exhibit their best performance with "Data"; this effect is examined further in Section 4.3). Search at evaluation time only provides a small boost in most environments, though it occasionally proves to be crucial, as in Acrobot and Go. Overall, using search during training ("Learn+Data") is the primary driver of MuZero's performance over the "One-Step" baseline, leveraging both better policy targets and an improved data distribution.

As discussed in Section 1, MuZero has important connections to a number of other approaches in MBRL; consequently, our results here suggest implications for some of these other methods. For example, the "Learn" variant of MuZero is similar to Dyna-style methods that perform policy updates with TRPO [34, 39, 46, 52]. Given the improved performance of "Learn+Data" over "Learn", these methods may similarly benefit from also using planning to select actions, and not just for learning. The "Data" variant of MuZero is similar in spirit to Dyna-2 [61]; our results suggest that works which implement this approach [5, 6] may similarly benefit from incorporating planning into policy and value updates. "Learn+Data+Eval" shows the contribution of MPC over "Learn+Data",

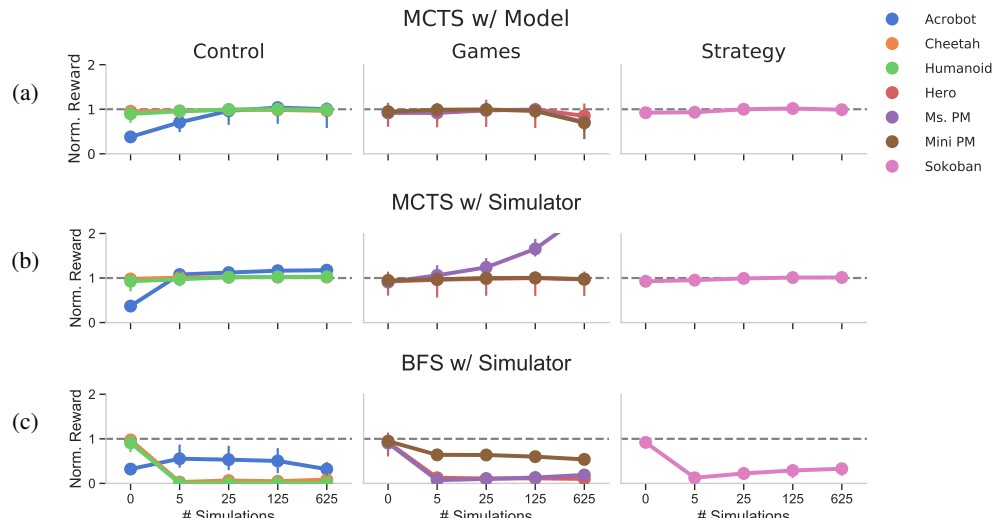

Figure 5: Effect of search at evaluation as a function of the number of simulations, normalized by the median scores in Table 10. All colored lines show medians across seeds, with error bars indicating min and max seeds. (a) MCTS with the learned model. (b) MCTS with the environment simulator. (c) Breadth-first search (BFS) with the environment simulator. Results with the learned model are similar and can be seen in Figure 11.

and combined with our results in Section 4.4, highlights the importance of guiding MPC with robust policy priors and/or value functions. Of course, one major difference between MuZero and all these other methods is that MuZero uses a value equivalent model which is not grounded in the original observation space [21]. We therefore reran some of our experiments using a model that is trained to reconstruct observations in order to measure any potential implications of this difference, but did not find that such models change the overall pattern of results (see Section D.2 and Figure 13-14).

## 4.3 PLANNING FOR LEARNING

**Tree depth** Figure 4a shows the result of varying tree depth $D_{\text{tree}} \in \{1, 2, 3, 5, \infty\}$ while keeping $D_{\text{UCT}} = D_{\text{tree}}$ and the search budget constant. Scores are normalized by the "Learn+Data" agent from Section 4.2. Strikingly, $D_{\text{tree}}$ does not make much of a difference in most environments. Even in Sokoban and Go, we can recover reasonable performance using $D_{\text{tree}} = 2$, suggesting that deep tree search may not be necessary for learning a strong policy prior, even in the most difficult reasoning domains. Looking at individual effects within each domain, we find that deep trees have a negative impact in Minipacman and an overall positive impact in Ms. Pacman, Acrobot, Sokoban, and Go (see Table 13). While we did not detect a quantitative effect in the other environments, qualitatively it appears as though very deep trees may cause worse performance in Hero.

**Exploration vs. exploitation depth** Figure 4b shows the strength of the policy prior as a result of manipulating the pUCT depth $D_{\text{UCT}} \in \{1, 2, 3, 6, \infty\}$ while keeping $D_{\text{tree}} = \infty$ and the search budget constant. Note that $D_{\text{UCT}} = 1$ corresponds to only exploring with pUCT at the root node and performing pure Monte-Carlo sampling thereafter. We find $D_{\text{UCT}}$ to have no effect in any environment except 9x9 Go (Table 14); Anthony et al. [4] also found larger values of $D_{\text{UCT}}$ to be important for Hex. Thus, exploration-exploitation deep within the search tree does not seem to matter at all except in the most challenging settings.

**Search budget** Figure 4c shows the strength of the policy prior after training with different numbers of simulations, with $D_{\text{tree}} = \infty$ and $D_{\text{UCT}} = 1$ (except in Go, where $D_{\text{UCT}} = \infty$), corresponding to exploring different actions at the root node and then performing Monte-Carlo rollouts thereafter. We opted for these values as they correspond to a simpler form of planning, and our previous experiments showed that larger settings of $D_{\text{UCT}}$ made little difference. We find an overall strong effect of the number of simulations on performance in all environments (Table 15). However, despite the overall positive effect of the search budget, too many simulations have a detrimental effect in many

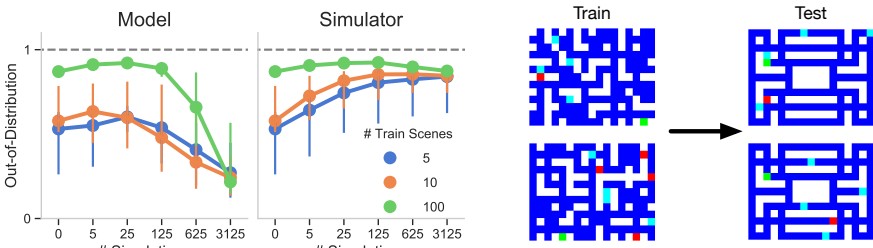

Figure 6: Generalization to out-of-distribution mazes in Minipacman. All points are medians across seeds (normalized by the median scores in Table 10), with error bars showing min and max seeds. Colors indicate agents trained on different numbers of unique mazes. The dotted lines indicate the baseline. The maps on the right give examples of the types of mazes seen during train and test. In-distribution generalization is shown in Figure 12 (Appendix) as the behavior is similar.

environments, replicating work showing that some amount of planning can be beneficial, but too much can harm performance [e.g., 34]. Additionally, the results with Ms. Pacman suggest that two simulations provide enough signal to learn well in some settings. It is possible that with further tuning, other environments might also learn effectively with smaller search budgets.

### 4.4 GENERALIZATION IN PLANNING

**Model generalization to new search budgets**  Figure 5a shows the results of evaluating the baseline agents (Section 4.1) using up to 625 simulations. As before, we find a small but significant improvement in performance of 7.4 percentage points between full MuZero and agents which do not use search at all ($t = -5.84, p < 0.001, N = 70$). Both Acrobot and Sokoban exhibit slightly better performance with more simulations, and although we did not perform experiments here with Go, Schrittwieser et al. [58] did and found a positive impact. However, Minipacman exhibits worse performance, and other environments show no overall effect (Table 16). The median reward obtained across environments at 625 simulations is also less than the baseline by a median of 4.7 percentage points ($t = -5.71, p < 0.001, N = 70$), possibly indicating an effect of compounding model errors. This suggests that for identical training and testing environments, additional search may not always be the best use of computation; we speculate that it might be more worthwhile simply to perform additional Dyna-like training on already-observed data [see "MuZero Reanalyze", 58].

**Policy and value generalization with a better model**  Planning with the simulator yields somewhat better results than planning with the learned model (Figure 5b), with all environments except Hero exhibiting positive rank correlations with the number of simulations (Table 17). Ms. Pacman, in particular, more than doubles in performance after 625 simulations[2]. However, across environments, 25, 125, and 625 simulations only increased performance over the baseline by a median of about 2 percentage points. Thus, planning with a better model may only produce small gains.

**Model generalization to new planners**  We find dramatic differences between MCTS and BFS, with BFS exhibiting a catastrophic drop in performance with any amount of search. This is true both when using the learned model (Figure 11, Appendix) and the simulator (Figure 5c), in which case the only learned component that is relied on is the value function. Consider the example of Sokoban, where the branching factor is five; therefore, five steps of BFS search means that all actions at the root node are expanded, and the action will be selected as the one with the highest value. Yet, the performance of this agent is substantially worse than just relying on the policy prior. This suggests a mismatch between the value function and the policy prior, where low-probability (off-policy) actions are more likely to have high value errors, thus causing problems when expanded by BFS. Moreover, this problem is not specific to BFS: in sparse reward environments, any planner (such as random shooting) that relies on the value function without the policy prior will suffer from this effect. This result highlights that in complex agent architectures involving multiple learned components, compounding error in the transition model is not the only source of error to be concerned about.

---

[2]However, using the simulator leaks information about the unobserved environment state, such as when the ghosts will stop being edible. Thus, these gains may overestimate what is achievable by a realistic model.

**Generalizing to new mazes** We trained Minipacman agents on 5, 10, or 100 unique mazes and then tested them on new mazes drawn either from the same distribution or a different distribution. Figure 6 shows the out-of-distribution results and Figure 12 the in-distribution results. Using the learned model, we see very slight gains in performance up to 125 simulations on both in-distribution and out-of-distribution mazes, with a sharp drop-off in performance after that reflecting compounding model errors in longer trajectories. The simulator allows for somewhat better performance, with greater improvements for small numbers of train mazes ($t = -10.43, p < 0.001, N = 360$, see also Table 18 and 19), indicating the ability of search to help with some amount of distribution shift when using an accurate model. However, as can be seen in the figure, this performance plateaus at a much lower value than what would be obtained by training the agent directly on the task. Moreover, reward using the simulator begins to decrease at 3125 simulations compared to at 125 simulations using the learned model ($t = -5.59, p < 0.001, N_1 = 20, N_2 = 20$), again indicating a sensitivity to errors in the value function and policy prior. In fact, this is the same effect as can be seen in Figure 5c; the only reason the drop-off in performance is less drastic than with BFS is because the policy prior used by MCTS keeps the search more on-policy.

## 5    Discussion

In this work, we explored the role of planning in MuZero [58] through a number of ablations and modifications. We sought to answer three questions: (1) In what ways does planning contribute to final performance? (2) What design choices within the planner contribute to stronger policy learning? (3) How well does planning support zero-shot generalization? In most environments, we find that (1) search is most useful in constructing targets for policy learning and for generating a more informative data distribution; (2) simpler and shallower planning is often as performant as more complex planning; and (3) search at evaluation time only slightly improves zero-shot generalization, and even then only if the model is highly accurate. Although all our experiments were performed with MuZero, these results have implications for other MBRL algorithms due to the many connections between MuZero and other approaches (Section 1).

A major takeaway from this work is that while search is useful for learning, simple and shallow forms of planning may be sufficient. This has important implications in terms of computational efficiency: the algorithm with $D_{\mathrm{UCT}} = 1$ can be implemented without trees and is thus far easier to parallelize than MCTS, and the algorithm with $D_{\mathrm{tree}} = 1$ can be implemented via model-free techniques [e.g., 1], suggesting that MBRL may not be necessary at all for strong final performance in some domains. Moreover, given that search seems to provide minimal improvements at evaluation in many standard RL environments, it may be computationally prudent to avoid using search altogether at test time.

The result that deep or complex planning is not always needed suggests that many popular environments used in MBRL may not be fully testing the ability of model-based agents (or RL agents in general) to perform sophisticated reasoning. This may be true even for environments which seem intuitively to require reasoning, such as Sokoban. Indeed, out of all our environments, only Acrobot and 9x9 Go strongly benefited from search at evaluation time. We therefore emphasize that for work which aims to build flexible and generalizable model-based agents, it is important to evaluate on a diverse range of settings that stress different types of reasoning.

Our generalization experiments pose a further puzzle for research on model-based reasoning. Even given a model with good generalization (e.g., the simulator), search in challenging environments is ineffective without a strong value function or policy to guide it. Indeed, our experiments demonstrate that if the value function and policy themselves do not generalize, then generalization to new settings will also suffer. But, if the value function and policy *do* generalize, then it is unclear whether a model is even needed for generalization. We suggest that identifying good inductive biases for policies which capture something about the world dynamics [e.g., 6, 22], as well as learning appropriate abstractions [30], may be as or more important than learning better models in driving generalization.

Overall, this work provides a new perspective on the contributions of search and planning in integrated MBRL agents like MuZero. We note that our analysis has been limited to single-task and (mostly) fully-observable, deterministic environments, and see similar studies focusing on multi-task and more strongly partially-observed and stochastic environments as important areas for future work.

ACKNOWLEDGMENTS

We are grateful to Ivo Danihelka, Michal Valko, Jean-bastien Grill, Eszter Vértes, Matt Overlan, Tobias Pfaff, David Silver, Nate Kushman, Yuval Tassa, Greg Farquhar, Loic Matthey, Andre Saraiva, Florent Altché, and many others for helpful comments and feedback on this project.

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

# A MuZero Algorithm Details

## A.1 MuZero pseudocode

Pseudocode for MuZero [58] is presented in Algorithm 1. Following initialization of the weights and the empty circular replay buffer $\mathcal{D}$, a learner and a number of actors execute in parallel, reading from and sending data to the replay buffer. In our experiments, the actors update their copies of the parameters $\theta$ after every 500 learner steps. Our setup and description follow that of Schrittwieser et al. [58].

---

**Algorithm 1** MuZero [58]

---

1: Initialize model $\mu_\theta$ and dataset $\mathcal{D}$

2: **function** ACTOR
3:     **while** true **do**
4:         $o \leftarrow$ initialize episode
5:         **while** episode not finished **do**
6:             $s \leftarrow h_\theta(\ldots, o)$
7:             $\pi^{\text{MCTS}}, v^{\text{MCTS}} \leftarrow \text{MCTS}(s, \mu_\theta)$     ▷ *Alternative: Monte-Carlo rollouts, BFS, etc.*
8:             $a \sim \pi^{\text{MCTS}}$                               ▷ *Alternative: sample action from $\pi_\theta$*
9:             $r^{\text{env}}, o' \leftarrow$ execute $a$ in environment
10:            Add $o, a, r^{\text{env}}, o', \pi^{\text{MCTS}}, v^{\text{MCTS}}$ to $\mathcal{D}$
11:            $o \leftarrow o'$
12:        **end while**
13:    **end while**
14: **end function**

15: **function** LEARNER
16:     **while** True **do**
17:         Sample batch of trajectories $B$ from $\mathcal{D}$
18:         $\ell \leftarrow$ compute loss (Equation 8) on $B$
19:         Update $\theta$ with gradient descent on $\ell$
20:     **end while**
21: **end function**

---

## A.2 Model details

In MuZero, the model $\mu_\theta$ is trained to directly predict three quantities for each future timestep $k = 1 \ldots K$. These are the policy $\pi^k_{\theta,t} \approx \pi^{\text{MCTS}}(a_{t+k}|o_1, \ldots, o_t, a_t, \ldots, a_{t+k-1})$, the value function $v^k_{\theta,t} \approx \mathbb{E}\left[r^{\text{env}}_{t+k+1} + \gamma r^{\text{env}}_{t+k+2} + \ldots | o_1, \ldots, o_t, a_t, \ldots, a_{t+k-1}\right]$, and the immediate reward $r^k_{\theta,t} \approx r^{\text{env}}_{t+k}$, where $\pi^{\text{MCTS}}$ is the policy used to select actions in the environment, $r^{\text{env}}$ is the observed reward, and $\gamma$ is the environment discount factor.

## A.3 MCTS details

Pseudocode for MCTS is presented in Algorithm 2. MuZero uses the pUCT rule [53] for the search policy. The pUCT rule maximizes an upper confidence bound [38] to balance exploration and exploitation during search. Specifically, the pUCT rule selects action

$$a^k = \arg\max_a \left[ Q(s,a) + \pi_\theta(s,a) \cdot \frac{\sqrt{\sum_b N(s,b)}}{1 + N(s,a)} \cdot \left( c_1 + \log\left( \frac{\sum_b N(s,b) + c_2 + 1}{c_2} \right) \right) \right], \tag{1}$$

where $N$ is the number of times $a$ has been selected during search, $Q$ is the average cumulative discounted reward of $a$, and $c_1, c_2$ are constants that control the relative influence of $Q$ and $\pi_\theta$. Following Schrittwieser et al. [58], we set $c_1 = 1.25$ and $c_2 = 19652$ in our experiments. At the root node alone, a small amount of Dirichlet noise is added to the policy prior $\pi_\theta$ to encourage additional exploration at the root.

---

**Algorithm 2** MCTS in MuZero

---

1: **function** MCTS(root state $s^0$, model $\mu_\theta$, number of simulations $B$)
2:     initialize edge statistics $\{(N(s^0, a), Q(s^0, a))\}_a$ to 0
3:     **for** $k = 1 \ldots B$ **do**
4:         $r^l, s^l, v^l \leftarrow$ SEARCHANDEXPAND($s^0$)
5:         BACKUP($r^l, s^l, v^l$)
6:     **end for**
7:     **return** $\pi^{\text{MCTS}}$ and $v^{\text{MCTS}}$ (Equation 6 and 7)
8: **end function**

9: **function** SEARCHANDEXPAND($s$)
10:     **while** true **do**
11:         $a \leftarrow$ PUCT($s$)                            ▷ *choose action according to pUCT*
12:         $r', s' \leftarrow$ TRANSITION($s, a$)                ▷ *use cached values if possible*
13:         **if** $N(s, a) = 0$ **then**
14:             add node $s'$ as child of $s$ on edge $a$
15:             initialize edge statistics $\{(N(s', a'), Q(s', a'))\}_{a'}$ to 0
16:             compute $\pi_\theta, v_\theta \leftarrow f_\theta(s')$
17:             **return** $r', s', v_\theta$
18:         **end if**
19:         $s \leftarrow s'$
20:     **end while**
21: **end function**

22: **function** BACKUP($r', s', v'$)
23:     **for** each edge on the path from $s'$ to the root $s^0$ **do**
24:         update statistics $N, Q$ with Equation 3 and 4
25:     **end for**
26: **end function**

---

Within the search tree, each node has an associated hidden state $s$. For each action $a$ from $s$ there is an edge $(s, a)$ on which the number of visits $N(s, a)$ and the current value estimate $Q(s, a)$ are stored. When a new node with state $s$ is created in the expansion step, the statistics for each edge are initialized as $\{N(s, a) = 0, Q(s, a) = 0\}$. The estimated policy prior $\pi_\theta(s, a)$, reward $r_\theta$, and state transition are also stored after being computed on expansion since they are deterministic and can be cached. Thus, the model only needs to be evaluated once per simulation when the new leaf is added.

When backing up after expansion, the statistics on all of the edges from the leaf to the root are updated. Specifically, for $k = l \ldots 0$, a bootstrapped $l - k$-step cumulative discounted reward estimate

$$G^k = \sum_{\tau=0}^{\ell-1-k} \gamma^\tau r_\theta^{k+1+\tau} + \gamma^{\ell-k} v_\theta^\ell, \tag{2}$$

is computed. The statistics for each edge $(s^k, a^k)$ for $k = 0 \ldots l - 1$ in the simulation path are updated as

$$Q(s^k, a^k) = \frac{N(s^k, a^k) \cdot Q(s^k, a^k) + G^k}{N(s^k, a^k) + 1} \tag{3}$$

$$N(s^k, a^k) = N(s^k, a^k) + 1. \tag{4}$$

To keep $Q$ estimates bounded within $[0, 1]$, the $Q$ estimates are first normalized as $\bar{Q} \in [0, 1]$ before passing them to the pUCT rule. The normalized estimates are computed as

$$\bar{Q}(s^k, a^k) = \frac{Q(s^k, a^k) - Q_{\min}}{Q_{\max} - Q_{\min}}, \tag{5}$$

where $Q_{\min} = \min_{(s,a)\in\text{Tree}} Q(s, a)$ and $Q_{\max} = \max_{(s,a)\in\text{Tree}} Q(s, a)$ are the minimum and maximum $Q$ values observed in the search tree so far.

After all simulations are complete, the policy $\pi^{\text{MCTS}}$ returned by MCTS is the visit count distribution at the root $s^0$ parameterized by a temperature $T$

$$\pi^{\text{MCTS}}(a) = \frac{N(s^0, a)^{1/T}}{\sum_b N(s^0, b)^{1/T}}. \tag{6}$$

During training, the temperature is set as a function of the number of learner update steps. Specifically, the temperature is set to 1 and then decayed by a factor of $0.95$ after every $5000$ steps.

The value $v^{\text{MCTS}}$ returned by MCTS is the average discounted return over all simulations

$$v^{\text{MCTS}} = \sum_a \left( \frac{N(s^0, a)}{\sum_b N(s^0, b)} \right) Q(s^0, a). \tag{7}$$

## A.4 TRAINING DETAILS

The MCTS policy is used to select an action $a_t \sim \pi_t^{\text{MCTS}}$, which is then executed in the environment and a reward $r_t^{\text{env}}$ observed. The model is jointly trained to match targets constructed from the observed rewards and the MCTS policy and value for each future timestep $k$. The policy targets are simply the MCTS policies, while the value targets are the $n$-step bootstrapped discounted returns $z_t = r_{t+1}^{\text{env}} + \gamma r_{t+2}^{\text{env}} + \cdots + \gamma^{n-1} r_{t+n}^{\text{env}} + \gamma^n v_{t+n}^{\text{MCTS}}$. For reward, value, and policy losses $\ell^r, \ell^v$, and $\ell^p$, respectively, the overall loss is then

$$\ell_t(\theta) = \sum_{k=0}^{K} \ell^r(r_{\theta,t}^k, r_{t+k}^{\text{env}}) + \ell^v(v_{\theta,t}^k, z_{t+k}) + \ell^p(\pi_{\theta,t}^k, \pi_{t+k}^{\text{MCTS}}) + c||\theta||^2, \tag{8}$$

where $c||\theta||^2$ is an L2 regularization term. For the rewards, values, and policies, a cross-entropy loss is used for each of $\ell^r, \ell^v$, and $\ell^p$.

# B FURTHER IMPLEMENTATION DETAILS

## B.1 DEPTH-LIMITED MCTS

The depth-limited MCTS algorithm is implemented by replacing the regular MCTS search and expand subroutines with a modified subroutine as follows. The backup subroutine remains unchanged.

---

**Algorithm 3** Search and expand subroutine for depth-limited MCTS.

---

1: **function** SEARCHANDEXPAND(root state $s^0$, max UCT depth $D_{\text{UCT}}$, max tree depth $D_{\text{tree}}$)
2:      $k \leftarrow 0$
3:      **while** $k < D_{\text{tree}}$ and $s^k$ is not a leaf **do**                      ▷ *Search for leaf node*
4:          $a \leftarrow$ SELECTACTION$(s^k, k, D_{\text{UCT}})$
5:          $s^{k+1} \leftarrow$ TRANSITION$(s^k, a)$
6:          $k \leftarrow k + 1$
7:      **end while**
8:      **if** $k < D_{\text{tree}}$ **then**                          ▷ *Expand unless maximum depth is reached*
9:          $a \leftarrow$ SELECTACTION$(s^k, k, D_{\text{UCT}})$
10:         $s^{k+1} \leftarrow$ TRANSITION$(s^k, a)$
11:         Add $s^{k+1}$ to tree
12:      **end if**
13:      **return** $s^{k+1}$
14: **end function**

15: **function** SELECTACTION$(s, k, D_{\text{UCT}})$                      ▷ *The search policy*
16:      **if** $k < D_{\text{UCT}}$ **then**
17:          $a \leftarrow$ PUCT$(s)$                      ▷ *Choose action according to pUCT*
18:      **else**
19:          $a \sim \pi_\theta(\cdot|s)$                      ▷ *Sample from prior*
20:      **end if**
21:      **return** a
22: **end function**

---

We initially tried varying $D_{\text{tree}}$ and $D_{\text{UCT}}$ together on Ms. Pacman and Minipacman. However, as we did not see any effect, we varied these variables separately for the remainder of our experiments in order to limit computation.

We did not run any experiments with $D_{\text{UCT}} = 0$, which corresponds to pure Monte-Carlo search. This is because this variant introduces a confound: with $D_{\text{UCT}} = 0$, the visit counts are no longer informative and unsuitable for use as policy learning targets. To test $D_{\text{UCT}} = 0$ would therefore also require modifying the learning target. Future work could test this by using the same MPO update described in the next section.

Additionally, we note that depth-limited MCTS will converge to the BFS solution (Section B.4) in the limit of infinite simulations. For finite simulations, depth-limited MCTS interpolates between the policy prior and the BFS policy.

## B.2 MuZero Variants

As described in Section 3, we implemented the following variants of MuZero:

- **One-Step**. This variant uses $D_{\text{tree}} = 1$ for learning, and acts by sampling from the policy prior. We also only train the model to predict one time step into the future (in all other variants, we train it to predict five time steps into the future). This variant is therefore as close as possible to a model-free version of MuZero, and could in principle be implemented solely with a Q-function.
- **Learn**. This variant uses $D_{\text{tree}} = \infty$ for learning, and acts by sampling from the policy prior. It therefore gets the benefits of a deep tree search for learning, but not for acting.
- **Data**. This variant uses $D_{\text{tree}} = 1$ for learning, acts during training by sampling from $\pi^{\text{MCTS}}$ (computed using $D_{\text{tree}} = \infty$), and acts at test time from the policy prior. This allows us to measure the impact of deep search on the distribution of data experienced during learning.
- **Learn+Data**. This variant using $D_{\text{tree}} = \infty$ both for learning and for acting during training; actions at test time are sampled from the policy prior. This variant thus benefits from search in two ways during learning, but uses no search at test time.
- **Learn+Data+Eval**. This variant is equivalent to the original version of MuZero, using search (with $D_{\text{tree}} = \infty$) for learning and for all action selection.

Another way to test the impact of search on learning and the data distribution would be to add a secondary model-free loss function to MuZero, making it easier to fully ablate the model-based loss function in the One-Step and Data variants. While we leave this exploration to future work, we mention that Hamrick et al. [27] investigated another member of the approximate policy iteration family which relies solely on Q-functions and which combines a model-free loss with a model-based loss. Thus, they were able to test the effect of a purely model-free variant, as well as a "Data"-like variant, both of which underperformed the full version of their model.

## B.3 Regularized policy updates (MPO) with MCTS

Past work has demonstrated that MuZero's policy targets suffer from degeneracies at low visit counts [27, 20]. To account for this, we modified the policy targets to use an MPO-style update [1] rather than the visit count distribution, similar to [20]: $\pi_{t+k}^{\text{MPO}} \propto \pi_{\theta,t}^k \cdot \exp\left(\mathbf{q}^{\text{MCTS}}/\tau\right)$. Here, $\mathbf{q}^{\text{MCTS}}$ are the Q-values at the root node of the search tree; $\tau = 0.1$ is a temperature parameter; and we use $\pi_{t+k}^{\text{MPO}}$ in place of $\pi_{t+k}^{\text{MCTS}}$ in Equation 8. Note that the Q-values for unvisited actions are set to zero; while it is in general a poor estimate for the true Q-function, we found this choice to outperform setting the Q-function for unvisited actions to the value function. This is likely because unvisited actions are unlikely under the prior and perhaps ought not be reinforced unless good estimates are obtained through exploration. However, this choice leads to a biased MPO update; how to unbias it will be a topic of further research. Similarly, we chose the MPO update for its ease of implementation, but it is likely other forms of regularized policy gradient (e.g. TRPO [59], or more generally natural or mirror policy optimization [72, 2]) would result in quantitively similar findings. We also did not tune $\tau$ for different environments; it is likely that properly tuning it could further improve performance of the agent at small search budgets.

### B.4 BREADTH-FIRST SEARCH

In our generalization experiments, we replaced the MCTS search with a breadth-first search algorithm. BFS explores all children at a particular depth of the tree in an arbitrary order before progressing deeper in the tree. Our implementation of BFS does not use the policy prior. Additionally, when performing backups, we compute the maximum over all values seen rather than averaging. Final actions are selected based on the highest Q-value after search, rather than highest visit count. Thus, this implementation of BFS is (1) maximally exploratory and (2) relies mostly on the value estimates $v_{\theta,t}$ (especially when the simulator is used instead of the learned model).

## C ENVIRONMENT AND ARCHITECTURE DETAILS

We evaluate on the following environments.

- **Minipacman**: a toy version of Ms. Pacman with ghosts that move stochastically. We modified the version introduced by [22] such that the maze is procedurally generated on each episode. See Section C.1 for details.

- **Hero** (Atari): a sparse reward, visually complex video game. The goal is to navigate a character through a mine, clearing cave-ins, destroying enemies, and rescuing trapped miners.

- **Ms. Pacman** (Atari): a fast-paced, visually complex video game. The goal is to control Pacman to eat all the "food" in a maze, while avoiding being eaten by ghosts.

- **Acrobot Sparse Swingup** (Control Suite): a low-dimensional, yet challenging control task with sparse rewards. The task is to balance upright an under-actuated double pendulum.

- **Cheetah Run** (Control Suite): a six-dimensional control task, where the goal is to control the joints of a "cheetah" character to make it run forward in a 2D plane.

- **Humanoid Stand** (Control Suite): a 21-dimensional control task, where the goal is to control the joints of a humanoid character to make it stand up.

- **Sokoban**: a difficult puzzle game that involves pushing boxes onto targets and in which incorrect moves can be unrecoverable [51].

- **9x9 Go**: an easier version of Go than the full 19x19 game, provided by [42]. Evaluation is reported against Pachi [7], with $10^4$ evaluations per move (strong amateur play).

We now provide specific details on the network architectures and hyperparameters used for each environment. Unless specified and for layers where it is appropriate, all layers in the networks use padding 'SAME' and stride 1, and convolutions are 2-D with $3 \times 3$ kernels. All networks consist of an encoder $h_\theta$, a recurrently-applied dynamics function $g_\theta$, and a prior function $f_\theta$.

For most environments, the encoder $h_\theta$ is a resnet composed of a number of segments. Each segment consists of a convolution, a layer norm, a number of residual blocks, and a ReLU. Each residual block contains a layer norm, a ReLU, a convolution, a layer norm, a ReLU, and a final convolution. The input of the residual block is then added to the output of its final convolution.

For most environments, the dynamics network has the same structure as the encoder, with a different number of segments and residual blocks.

Finally, the prior function predicts three quantities and is composed of three separate networks: a policy head, a value head, and a reward head. The policy, value, and reward heads each consist of a $1 \times 1$ convolution followed by a number of linear layers, with a ReLU between each pair of layers.

The hyperparameters in Table 1 are shared across all environments except Go.

### C.1 MINIPACMAN

Minipacman is a toy version of Ms. Pacman. Unlike the Atari game, Minipacman also exhibits a small amount of stochasticity in that the ghosts move using an epsilon-greedy policy towards the agent.

Table 1: Shared hyperparameters

| Hyperparameters | Value |
| --- | --- |
| Dirichlet alpha | 0.3 |
| Exploration fraction | 0.25 |
| Exploration temperature | 1.0 |
| Temperature decay schedule | $5 \times 10^3$ |
| Temperature decay rate | 0.95 |
| Replay capacity | $5 \times 10^5$ |
| Min replay | $10^5$ |
| Sequence length | 5 |

### C.1.1 MAZES

For Minipacman, we altered the environment to support the use of both procedurally generated mazes ("in-distribution" mazes) and the standard maze ("out-of-distribution" mazes), both of size 15x19. Figure 6 shows example mazes of both types. In all experiments except generalization, we trained agents on an unlimited number of the "in-distribution" mazes, and tested on other mazes also drawn from this set. For the generalization experiments, we trained agents on a fixed set of either 5, 10, or 100 "in-distribution" mazes. Then, at evaluation time, we either tested on mazes drawn from the full in-distribution set or from the out-of-distribution set.

To generate the procedural mazes, we first used Prim's algorithm to generate corridors. To make the maze more navigable, we then randomly removed walls with a probability of $p = 0.3$. The number of initial ghosts was sampled as $g_0 \sim 1 + \text{Poisson}(1)$ and this number increased by $g_\Delta \sim 0.25 + U(0, 1)$ ghosts per level, such that the total number of ghosts at level $l$ was $g_l = \lfloor g_0 + (l - 1)g_\Delta \rfloor$. The number of pills was always set to 4. The default Minipacman maze was hand crafted to be similar to the Ms. Pacman maze. In the default maze, there is always one initial ghost ($g_0 = 1$) and this number increases by 1 every two levels ($g_\Delta = 0.5$). We similarly set the number of pills to 4. In all mazes, the initial locations of the ghosts, pills, and Pacman is randomly chosen at the beginning of every episode.

### C.1.2 NETWORK ARCHITECTURE

For Minipacman, the encoder has 2 segments, each with 64 channels and 2 residual blocks. The dynamics function has 1 segment with 5 residual blocks, all with 64 channels. The policy head has a convolution with 4 channels and one linear layer with a channel per action (in Minipacman, this is 5). The value head has a convolution with 32 channels and two linear layers with 64 and 601 channels. The reward network has the same structure as the value head.

All Minipacman experiments were run using 400 CPU-based actors and 1 NVIDIA V100 for the learner.

Table 2: Hyperparameters for Minipacman

| Hyperparameters | Value |
| --- | --- |
| Learning rate | $10^{-3}$ |
| Discount factor | 0.97 |
| Batch size | 512 |
| $n$-step return length | 10 |
| Replay samples per insert ratio | 0.25 |
| Learner steps | $2 \times 10^5$ |
| Policy loss weight | 1. |
| Value loss weight | 0.3 |
| Num simulations | 10 |
| Max steps per episode | 600 |

## C.2 ATARI

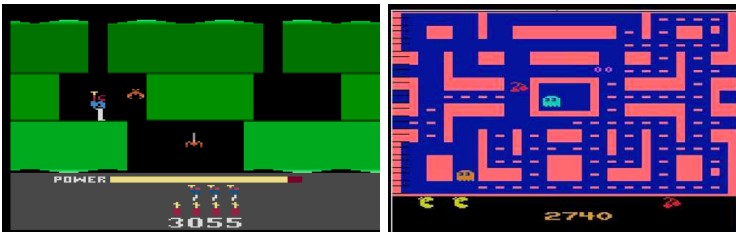

Figure 7: (*Left*) Hero, (*Right*) Ms.Pacman

The Atari Learning Environment [8] is a challenging benchmark of 57 classic Atari 2600 games played from pixel observations. We evaluate on Ms. Pacman and Hero. Each observation consists of the 4 previous frames and action repeats is 4. Note that, as MuZero does not incorporate recurrence, this limited number of frames makes some aspects of the the environment partially observable (such as when the ghosts turn from being edible to inedible).

For Atari, the encoder consists of 4 segments with $(64, 128, 128, 128)$ channels, each with 2 residual blocks, followed by 1 segment with 5 residual blocks with 128 channels. The dynamics network has 1 segment with 5 residual blocks with 128 channels. The heads are the same as in Minipacman.

All Atari experiments were run using 1024 CPU-based actors and 4 NVIDIA V100s for the learner.

Table 3: Hyperparameters for Atari

| Hyperparameters | Value |
|---|---|
| Learning rate | $10^{-3}$ |
| Discount factor | 0.995 |
| Batch size | 2048 |
| $n$-step return length | 10 |
| Replay samples per insert ratio | 0.25 |
| Learner steps | $1.5 \times 10^5$ |
| Policy loss weight | 1. |
| Value loss weight | 0.3 |
| Num simulations | 50 |

## C.3 CONTROL SUITE

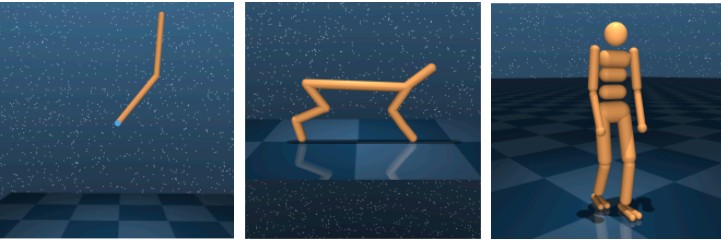

Figure 8: Control Suite environment: *Left* Acrobot *Middle* Cheetah *Right* Humanoid.

The DeepMind Control Suite [70] is a widely used benchmark for control tasks in MuJoCo. We select 3 high-dimensional environments Cheetah (Run), Acrobot (Swing-up Sparse) and Humanoid (Stand). We use the raw state observation as input and follow the same procedure as [20, 69] to discretize the continuous action space. Each action dimension is discretized into 5 bins evenly spaced between $[-1, 1]$.

For Control Suite environments, we use a model based on that of [44]. The encoder consists of a linear layer with 300 channels, a layer norm, a tanh, a linear layer with 200 channels, and an

exponential linear unit (ELU). The dynamics function is simply a linear layer with 200 channels followed by an ELU. The weights in the encoder and dynamics function are initialized uniformly. The policy head is a factored policy head composed of a linear layer with # dimensions × # bins channels. The factored policy head independently chooses an action for each dimension. The value and reward heads are both composed of a linear layer with 64 channels followed by a ReLU and then a linear layer with 2001 channels.

All Control Suite experiments were run using 1024 CPU-based actors and 2 second-generation (v2) Tensor Processing Units (TPUs) for the learner.

Table 4: Hyperparameters for control suite.

| Hyperparameters | Cheetah (Run) | Acrobot (Swing-up Sparse) | Humanoid (Stand) |
|---|---|---|---|
| Learning rate | $5 \times 10^{-4}$ | $2.5 \times 10^{-4}$ | $2.5 \times 10^{-4}$ |
| Discount factor | 0.995 | 0.995 | 0.995 |
| Batch size | 1024 | 1024 | 1024 |
| $n$-step return length | 50 | 50 | 30 |
| Replay samples per insert ratio | 25. | 2. | 15. |
| Learner steps | $5 \times 10^{6}$ | $5 \times 10^{6}$ | $5 \times 10^{6}$ |
| Policy loss weight | 1. | 1. | 1. |
| Value loss weight | 0.5 | 0.5 | 0.5 |
| Num simulations | 50 | 50 | 50 |

## C.4 SOKOBAN

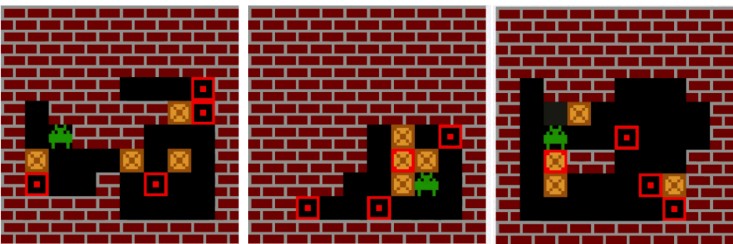

Figure 9: Sokoban environment.

Sokoban [51] is a classic puzzle problem, where the agent's task is to push a number of boxes onto target locations. In this environment many moves are irreversible as the boxes can only be pushed forward and hence the puzzle can become unsolvable if wrong moves are made.

For Sokoban, the encoder is the same as that used for Minipacman with an additional 256-channel $1 \times 1$ convolutional layer at the end. Instead of a resnet, for the dynamics network for Sokoban we used a DRC(3, 1) convolutional LSTM [22]. For the heads, we use the same network as in Minipacman except that the convolutional layers in the policy, value, and reward heads have 32, 32, and 16 channels, respectively.

All Sokoban experiments were run using 2048 CPU-based actors and 4 NVIDIA V100s for the learner.

## C.5 9x9 GO

9x9 Go is a smaller version of the full 19x19 game. While the board is fully observed, the actions of the other player cannot be fully predicted, thus making the game stochastic from each players' perspective.

For the Go experiments, we used a different implementation of the MuZero algorithm due to easier interfacing with the Go environment. The main difference between the implementation used for other environments and the one for Go is the data pipeline. In the first implementation, actors and

Table 5: Hyperparameters for Sokoban

| Hyperparameters | Value |
|---|---|
| Learning rate | $10^{-3}$ |
| Discount factor | 0.99 |
| Batch size | 2048 |
| $n$-step return length | 10 |
| Replay samples per insert ratio | 0.4 |
| Learner steps | $3 \times 10^5$ |
| Policy loss weight | 1. |
| Value loss weight | 0.3 |
| Num simulations | 25 |

learner communicate asynchronously through a replay buffer. In the one used for Go, there is no replay buffer; instead, actors add their data to a queue which the learner then consumes. Additionally, while the implementation of MuZero used in the other experiments trained the value function using $n$-step returns, $z_t = r_{t+1}^{\text{env}} + \gamma r_{t+2}^{\text{env}} + \cdots + \gamma^{n-1} r_{t+n}^{\text{env}} + \gamma^n v_{t+n}^{\text{MCTS}}$ (see Section 2), the one used here uses lambda returns, $z_t^\lambda = (1 - \lambda) \sum_{n=1}^{\infty} \lambda^{n-1} z_{t:t+n}$.

The two player aspect of the game is handled entirely by having a single player playing both moves, but using a discount of $-1$. The input to the agent is the last two states of the Go board and one color plane, where each state is encoded relative to color of the player with 3 planes, own stones, opponent's stones and empty stones.

The encoder consists of a single convolution layer with 128 channels, followed by 6 residual $3 \times 3$ convolutional blocks with 128 channels (each block consists of two convolutional layers with a skip connection). The network is size-preserving so hidden states are of size 9x9. The transition model takes as input the last hidden state and the action encoded as a one-hot plane, and also consists in 6 residual convolutional blocks with 128 channels. The representation model consists in one $1 \times 1$ convolutional layer with 2 channels, followed by an MLP with a single layer of 256 units.

Table 6: Hyperparameters for Go

| Hyperparameters | Value |
|---|---|
| Learning rate | $4 \times 10^{-4}$ |
| Discount factor | $-1$ |
| Batch size | 16 |
| $\lambda$ | 0.99 |
| Learner steps | $10^5$ |
| Policy loss weight | 1.0 |
| Value loss weight | 0.25 |
| Num simulations | 150 |
| Dirichlet alpha | 0.25 |
| Exploration fraction | 0.4 |

# D  ADDITIONAL RESULTS AND ANALYSIS

## D.1  EXTENSIONS TO FIGURES IN MAIN PAPER

Figure 10 shows the same information as Figure 3 (contributions of search to performance) but split into separate groups and with error bars shown. Figure 11 shows the same information as Figure 5 (effect of search at evaluation as a function of the number of simulations) but using breadth-first search with a learned model. Figure 12 shows the same information as Figure 6 (effect of search on generalization to new mazes) but for the in-distribution mazes instead of the out-of-distribution

mazes. Figure 22 presents learning curves for Go for different values of $D_{\mathrm{UCT}}$ and numbers of simulations.

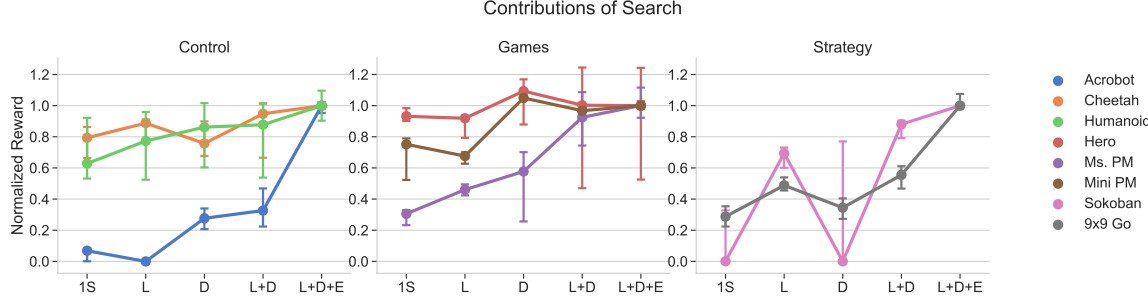

Figure 10: Contributions of the use of planning to performance. A breakdown containing the same information as Figure 3 with error bars showing the maximum and minimum seeds.

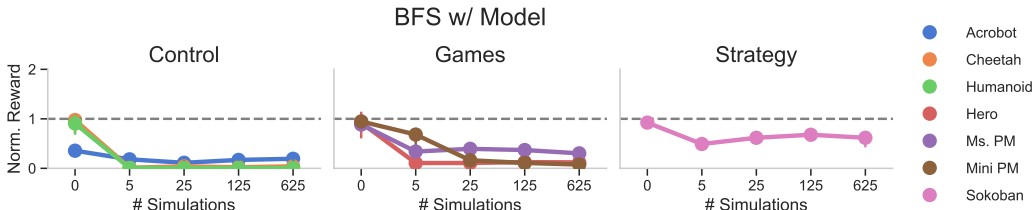

Figure 11: Effect of search at evaluation as a function of the number of simulations for breadth-first search (BFS) with the learned model. All colored lines show medians across seeds, with error bars indicating min and max seeds.

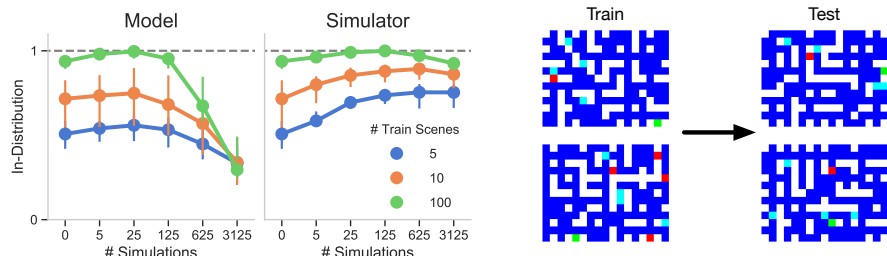

Figure 12: Effect of search on generalization to new in-distribution mazes in Minipacman. All points are medians across seeds, with error bars showing min and max seeds. Colors indicate agents trained on different numbers of unique mazes. The dotted lines indicate equivalent performance to the baseline. The maps on the right give examples of the types of mazes seen during train and test.

## D.2 MuZero with observation-reconstruction loss

As discussed above, planning in MuZero is performed entirely in a hidden space. This hidden space has no direct semantic association with the environment state as its sole purpose is to enable prediction of future rewards, policies, and values. This is in contrast, however, to much other work on model-based reinforcement learning, for which the model explicitly predicts environment states or observations. To investigate the effect of tying the model to the environment, we added an observation-reconstruction loss to our MuZero implementation and re-ran a subset of the experiments from Section 4.2 using this loss.

### D.2.1 OVERALL CONTRIBUTIONS OF PLANNING WITH RECONSTRUCTION LOSS

For two pixel-based environments (Minipacman and Sokoban), we added a binary cross-entropy loss between the true pixel observations and predicted future observations. Similarly, for the three state-based environments (Acrobot, Cheetah, and Humanoid), we added an L2 loss between the true state observations and the predicted future states. We evaluated the effect of the reconstruction losses on these environments, for each of the "Learn," "Learn+Data," and "Learn+Data+Eval" MuZero variants of Section 4.2. Final performance results are shown in Figure 13 and learning curves are shown in Figure 14.

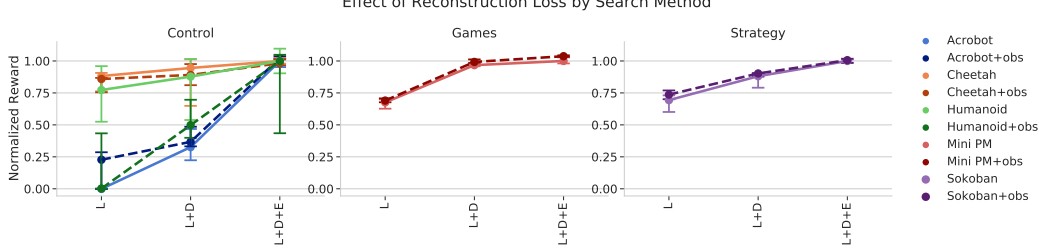

Figure 13: Contributions of planning to the performance of the "Learn (L)", "Learn+Data (L+D)", and "Learn+Data+Eval (L+D+E)" variants when training with and without an observation-reconstruction loss. Variants trained with the reconstruction loss are denoted "+obs" in the legend. Variants without the reconstruction loss are the same as those shown in Figure 10.

As shown by these figures, the performance of MuZero is relatively unchanged when using a reconstruction loss. The two main effects of the reconstruction loss, as shown by these figures, are (1) that learning consistently takes off faster when using the reconstruction loss (except in Humanoid) and (2) that the reconstruction loss allows the "Learn" variant of Acrobot to obtain reward. This latter point indicates that the model learned by "Learn" without the reconstruction loss is impoverished in a way that causes compounding errors with forward credit assignment, as discussed in van Hasselt et al. [73]. Overall, our results indicate that learning a model in observation space does not fundamentally alter the role of planning in MuZero but it does improve the fidelity of the model, which can benefit planning. Given the connections between MuZero and MBRL in general, we postulate that this conclusion holds true for other MBRL methods as well.

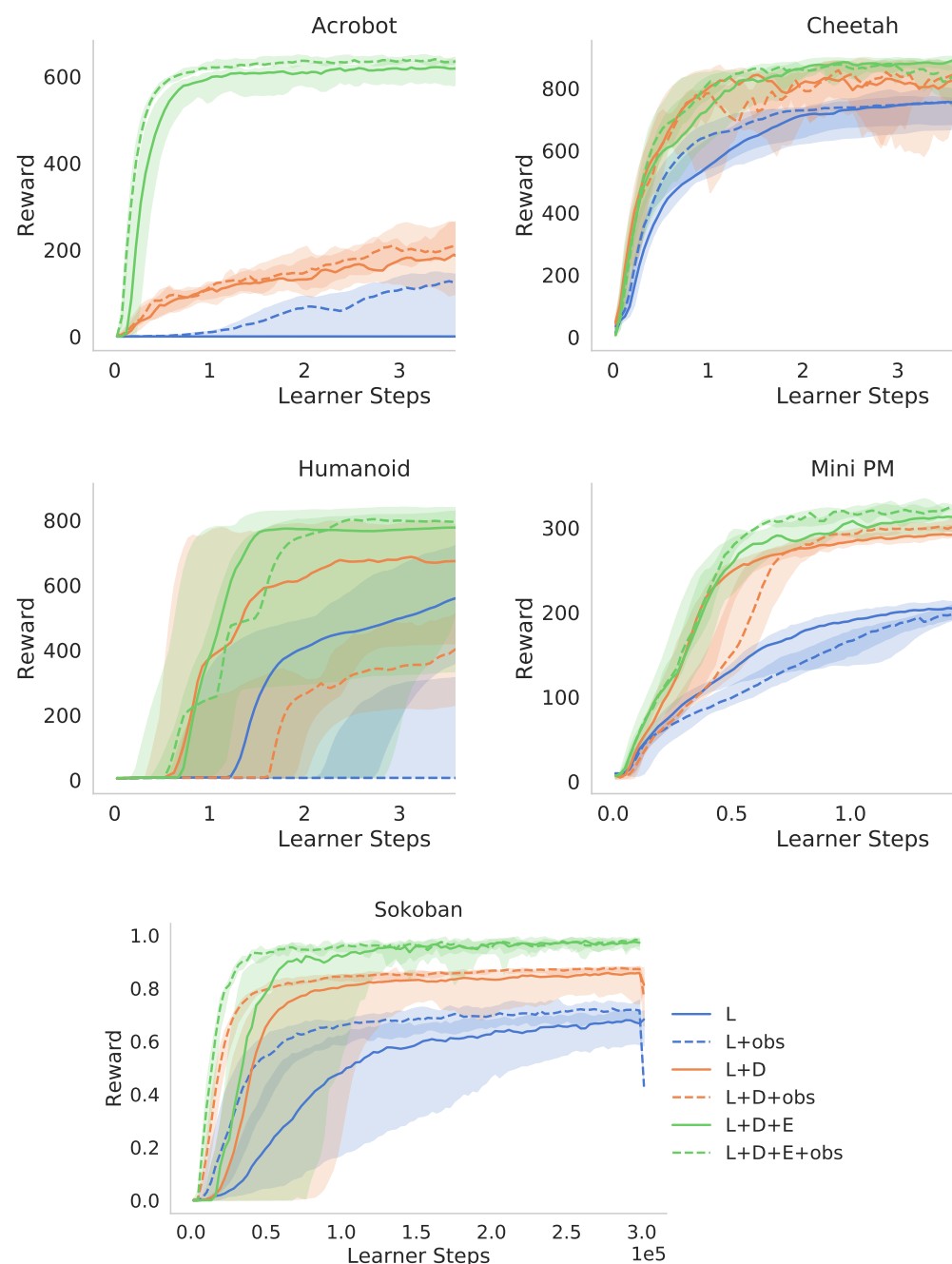

Figure 14: Learning curves for the "Learn (L)", "Learn+Data (L+D)", and "Learn+Data+Eval (L+D+E)" variants when training with and without an observation-reconstruction loss. Variants trained with the reconstruction loss are denoted "+obs" in the legend. The variants without the reconstruction loss are the same as those shown in Section D.3.

### D.2.2 OBSERVATION-RECONSTRUCTION LOSS IMPLEMENTATION DETAILS

To implement the reconstruction loss, we added a reconstruction function $\hat{o}_{\theta,t}^k = \rho_\theta(s_t^k)$ to the model $\mu_\theta$ that takes in the hidden state $s_t$ and generates a predicted observation $\hat{o}_{\theta,t}^k$ for the $k$th imagined step after timestep $t$. The predicted observation is then fed into an additional term $\ell^\rho(\hat{o}_t^k, o_{t+k})$ in

the MuZero loss (Equation 8). The parameters and architectures were lightly tuned on a subset of these environments.

For the pixel-based environments (i.e., Minipacman and Sokoban), $\rho_\theta$ is a simple 3-layer convolutional network with $3 \times 3$ kernels, 128 channels per layer, and ReLU activations feeding into a linear layer with 3 channels and a sigmoid activation. The loss $\ell^\rho$ is the binary cross-entropy loss.

Similarly, for the state-based environments (i.e., Acrobot, Cheetah, and Humanoid), $\rho_\theta$ is a simple feed-forward network with 3 linear layers of size $[200, 100, |O|]$ and ELU activations, where $|O|$ is the size of the state vector. The output of this is fed into $\ell^\rho$, which is an L2 loss.

### D.3 BASELINE LEARNING CURVES

Figures 15, 16 17, 18, 19, 20, 21, 22 show the learning curves for each environment for each experiment in the main body. In all plots, shaded regions indicate minimum and maximum seeds, while lines show the median across seeds.

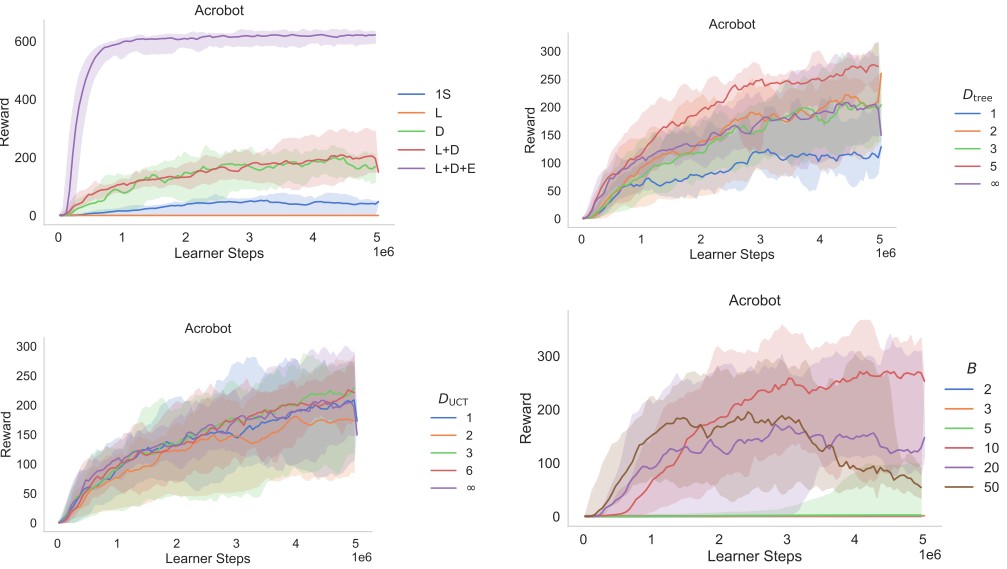

Figure 15: Learning curves for Acrobot.

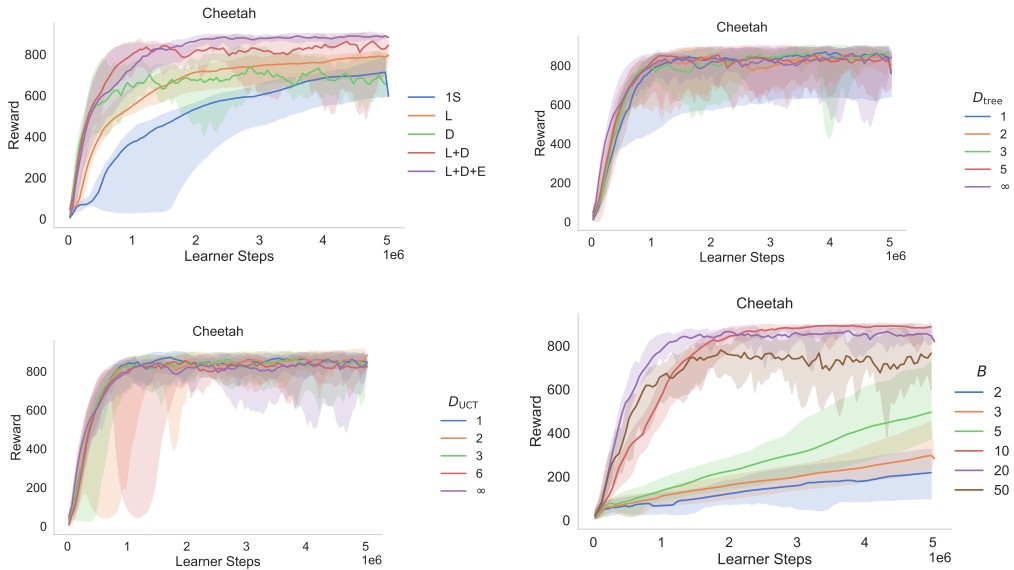

Figure 16: Learning curves for Cheetah.

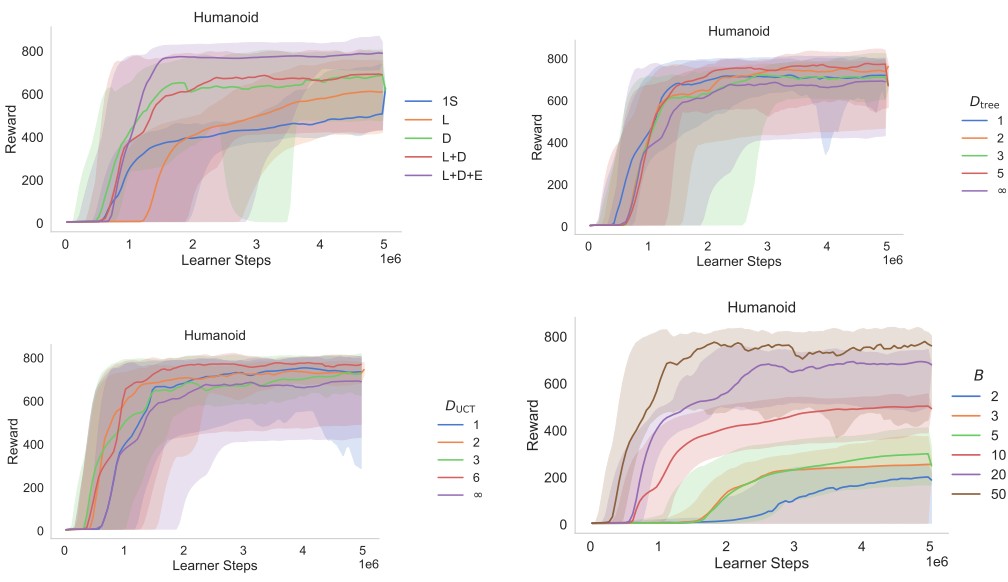

Figure 17: Learning curves for Humanoid.

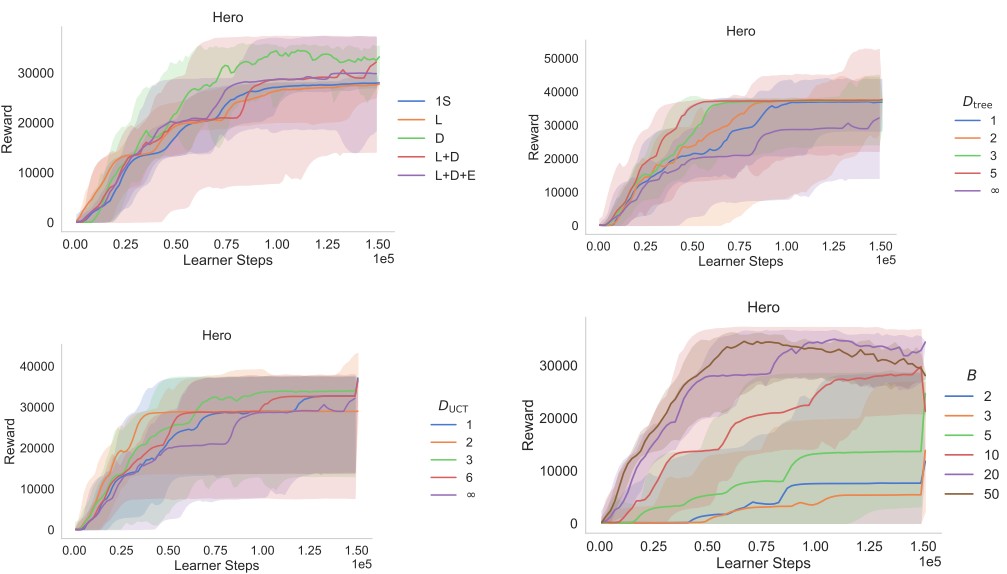

Figure 18: Learning curves for Hero.

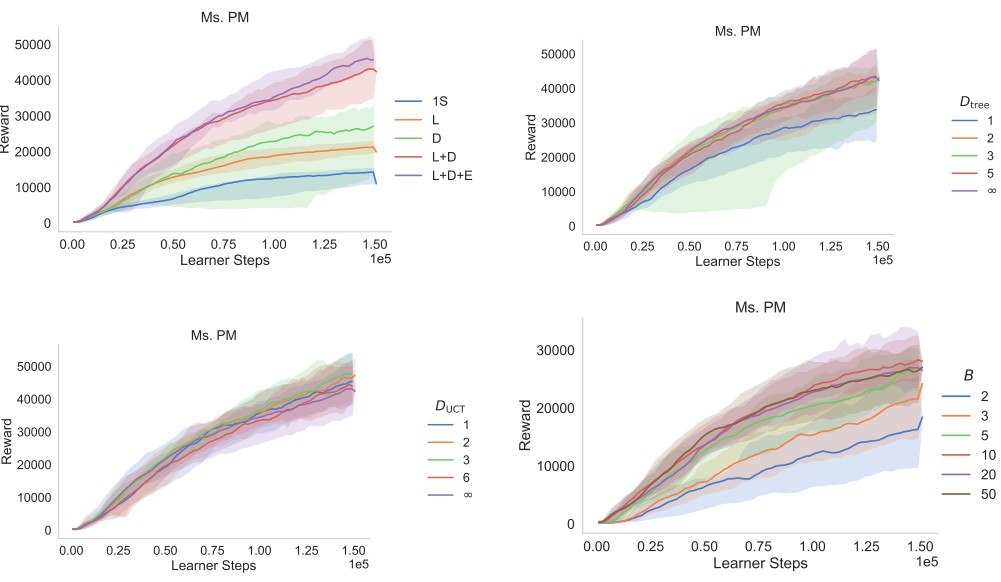

Figure 19: Learning curves for Ms. Pacman.

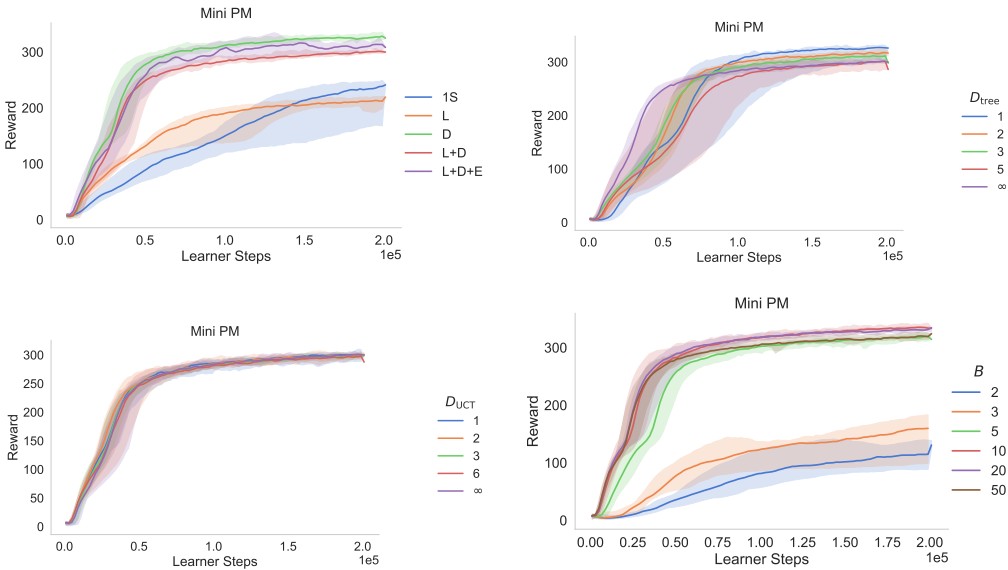

Figure 20: Learning curves for Minipacman.

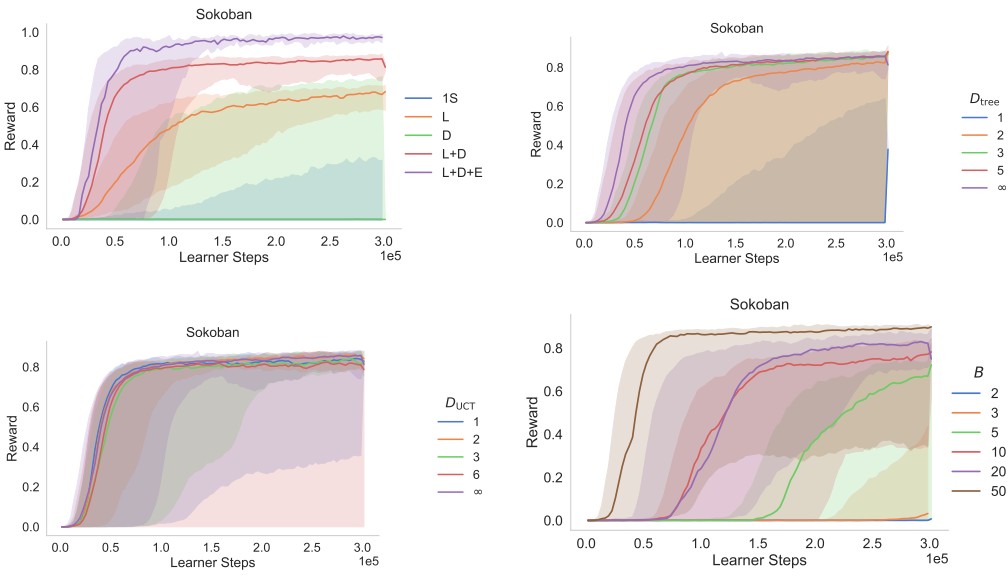

Figure 21: Learning curves for Sokoban.

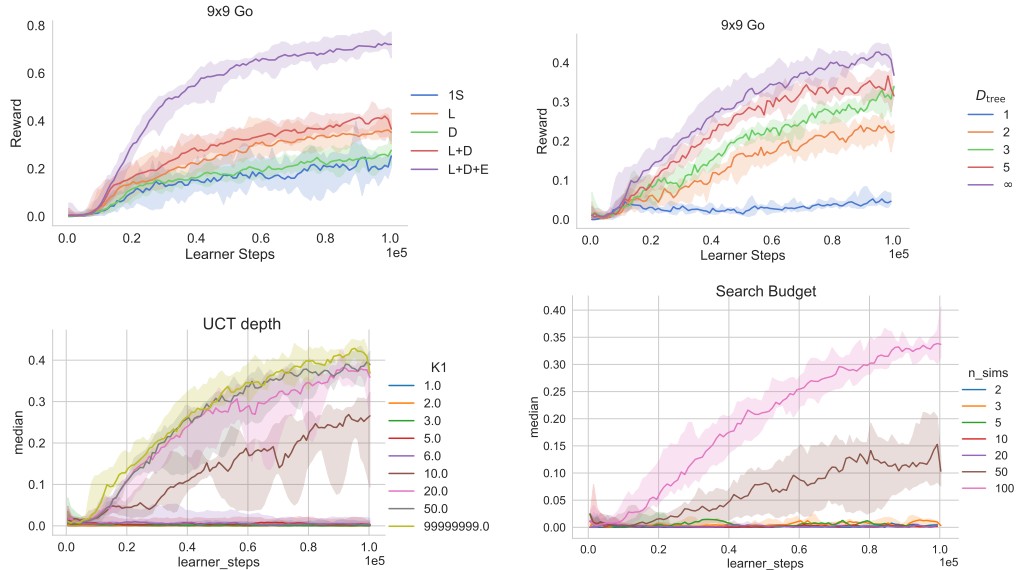

Figure 22: Learning curves for Go, including extra experiments on search budget and $D_{\text{UCT}}$.

## D.4    BASELINE VALUES

Table 7:    Values obtained by the baseline vanilla MuZero agent (corresponding to the "Learn+Data+Eval" agent in Figure 3), computed from the average of the last 10% of scores seen during training. Shown are the median across ten seeds, as well as the worst and best seeds. Median values are used to normalize the results in Figure 3.

|  | **Median** | **Worst Seed** | **Best Seed** |
|---|---|---|---|
| Acrobot | 620.07 | 590.66 | 632.83 |
| Cheetah | 885.57 | 868.04 | 904.64 |
| Humanoid | 787.98 | 712.33 | 863.27 |
| Hero | 29916.97 | 16104.65 | 37180.81 |
| Ms. Pacman | 45346.71 | 41840.45 | 50584.15 |
| Minipacman | 310.1 | 304.31 | 318.58 |
| Sokoban | 0.97 | 0.96 | 0.98 |
| 9x9 Go | 0.72 | 0.7 | 0.77 |

Table 8: Values obtained by MuZero at the very start of training (i.e., with a randomly initialized policy). Values are computed from the average of the first 1% of scores seen during training. Shown are the median across ten seeds, as well as the worst and best seeds. Median values are used to normalize the results in Figure 3.

|  | **Median** | **Worst Seed** | **Best Seed** |
|---|---|---|---|
| Acrobot | 0.54 | 0.17 | 1.26 |
| Cheetah | 6.48 | 2.6 | 42.54 |
| Humanoid | 5.36 | 3.22 | 9.92 |
| Hero | 7.98 | 0.0 | 364.96 |
| Ms. Pacman | 247.93 | 169.99 | 315.68 |
| Minipacman | 6.8 | 3.71 | 10.4 |
| Sokoban | 0.0 | 0.0 | 0.0 |
| 9x9 Go | 0.0 | 0.0 | 0.06 |

Table 9: Values obtained by a version of MuZero that uses no search at evaluation time (corresponding to the "Learn+Data" agent in Figure 3). Shown are the median across ten seeds, as well as the worst and best seeds. Median values are used to normalize the results in Figure 4.

|  | **Median** | **Worst Seed** | **Best Seed** |
| --- | --- | --- | --- |
| Acrobot | 202.66 | 139.01 | 290.64 |
| Cheetah | 839.59 | 591.52 | 896.61 |
| Humanoid | 692.05 | 426.25 | 799.85 |
| Hero | 29970.25 | 14059.2 | 37245.22 |
| Ms. Pacman | 41959.6 | 33780.17 | 49255.98 |
| Minipacman | 300.25 | 297.61 | 302.36 |
| Sokoban | 0.85 | 0.77 | 0.88 |
| 9x9 Go | 0.4 | 0.34 | 0.44 |

Table 10: Values obtained by a baseline vanilla MuZero agent, evaluated offline from a checkpoint saved at the very end of training. For each seed, values are the average over 50 (control tasks and Atari) or 1000 episodes (Minipacman and Sokoban). These values are used to normalize the results in Figure 5 and Figure 6. Note that for Minipacman, the scores reported here are for agents that were both trained and tested on either the in-distribution mazes or the out-of-distribution mazes. Shown are the median across ten seeds, as well as the worst and best seeds.

|  | **Median** | **Worst Seed** | **Best Seed** |
| --- | --- | --- | --- |
| Acrobot | 558.18 | 366.84 | 625.72 |
| Cheetah | 896.56 | 806.24 | 905.91 |
| Humanoid | 790.8 | 704.95 | 867.8 |
| Hero | 32545.4 | 19350.8 | 37234.2 |
| Ms. Pacman | 45145.0 | 42776.6 | 54206.4 |
| Minipacman (In-Distribution) | 319.81 | 312.87 | 331.37 |
| Minipacman (Out-of-Distribution) | 498.68 | 494.05 | 504.19 |
| Sokoban | 0.96 | 0.94 | 0.97 |

## D.5 OVERALL CONTRIBUTIONS OF PLANNING

Table 11: Values in Figure 3. Each column shows scores where 0 corresponds to the reward obtained by a randomly initialized agent (Table 8) and 100 corresponds to full MuZero ("Learn+Data+Eval", Table 7).

|            | Learn | Data  | Learn+Data |
|------------|-------|-------|------------|
| Acrobot    | -0.1  | 27.7  | 32.6       |
| Cheetah    | 88.8  | 75.7  | 94.8       |
| Humanoid   | 77.3  | 86.2  | 87.7       |
| Hero       | 91.9  | 109.4 | 100.3      |
| Ms. Pacman | 46.0  | 57.7  | 92.5       |
| Minipacman | 67.6  | 104.9 | 96.8       |
| Sokoban    | 69.4  | 0.0   | 88.1       |
| 9x9 Go     | 48.8  | 34.5  | 55.5       |
| Median     | 68.5  | 66.7  | 90.3       |

Table 12: Effect of the different contributions of search, modeled as `Reward ~ Environment + TrainUpdate * TrainAct + TestAct` over $N = 400$ data points, using the levels for each variable as defined in the table in Figure 3. This ANOVA indicates that both the environment, model-based learning, model-based acting during training, and model-based acting during testing are all significant predictors of reward. We did not detect an interaction between model-based learning and model-based acting during learning.

| Variable            | Statistic                                              | Strength of Evidence |
|---------------------|-------------------------------------------------------|----------------------|
| Environment         | $F(7, 388) = 70.95, p < 0.001$                        | ***                  |
| TrainUpdate         | $F(1, 388) = 175.93, p < 0.001$                       | ***                  |
| TrainAct            | $F(1, 388) = 146.73, p < 0.001$                       | ***                  |
| TestAct             | $F(1, 388) = 59.32, p < 0.001$                        | ***                  |
| TrainUpdate:TrainAct| $F(1, 388) = 1.15, p = 0.28$                          |                      |
| TrainUpdate (MB - MF)| $t = 4.42, p < 0.001, N_1 = 240, N_2 = 160$          | ***                  |
| TrainAct (MB - MF)  | $t = 5.11, p < 0.001, N_1 = 240, N_2 = 160$           | ***                  |
| TestAct (MB - MF)   | $t = 6.83, p < 0.001, N_1 = 80, N_2 = 320$            | ***                  |

## D.6 PLANNING FOR LEARNING

Table 13: Effect of tree depth, $D_{\text{tree}}$, modeled as `Reward ~ Environment * log($D_{\text{tree}}$)` over $N = 375$ data points. Where $D_{\text{tree}} = \infty$, we used the value for the maximum possible depth (i.e. the search budget). Top: this ANOVA indicates that both the environment and tree depth are significant predictors of reward, and that there is an interaction between environment and tree depth. Bottom: individual Spearman rank correlations between reward and $\log(D_{\text{tree}})$ for each environment. $p$-values are adjusted for multiple comparisons using the Bonferroni correction.

| Variable | Statistic | Strength of Evidence |
|---|---|---|
| Environment | $F(7, 359) = 25.23, p < 0.001$ | *** |
| $\log(D_{\text{tree}})$ | $F(1, 359) = 22.95, p < 0.001$ | *** |
| Environment:$\log(D_{\text{tree}})$ | $F(7, 359) = 12.99, p < 0.001$ | *** |
| Acrobot | $\rho = 0.55, p < 0.001, N = 50$ | *** |
| Cheetah | $\rho = -0.14, p = 1.00, N = 50$ | |
| Humanoid | $\rho = -0.01, p = 1.00, N = 50$ | |
| Hero | $\rho = -0.27, p = 0.49, N = 50$ | |
| Ms. Pacman | $\rho = 0.52, p < 0.001, N = 50$ | *** |
| Minipacman | $\rho = -0.88, p < 0.001, N = 50$ | *** |
| Sokoban | $\rho = 0.59, p < 0.001, N = 50$ | *** |
| 9x9 Go | $\rho = 0.96, p < 0.001, N = 25$ | *** |

Table 14: Effect of exploration vs. exploitation depth, $D_{\text{UCT}}$, modeled as `Reward ~ Environment * log($D_{\text{UCT}}$)` over $N = 375$ data points. Where $D_{\text{UCT}} = \infty$, we used the value for the maximum possible depth (i.e. the search budget). Top: this ANOVA indicates that neither the environment nor exploration vs. exploitation depth are significant predictors of reward. Bottom: individual Spearman rank correlations between reward and $\log(D_{\text{UCT}})$ for each environment. $p$-values are adjusted for multiple comparisons using the Bonferroni correction. The main effects are primarily driven by Go.

| Variable | Statistic | Strength of Evidence |
|---|---|---|
| Environment | $F(7, 359) = 71.96, p < 0.001$ | *** |
| $\log(D_{\text{UCT}})$ | $F(1, 359) = 13.06, p < 0.001$ | *** |
| Environment:$\log(D_{\text{UCT}})$ | $F(7, 359) = 15.55, p < 0.001$ | *** |
| Acrobot | $\rho = 0.15, p = 1.00, N = 50$ | |
| Cheetah | $\rho = -0.25, p = 0.65, N = 50$ | |
| Humanoid | $\rho = -0.05, p = 1.00, N = 50$ | |
| Hero | $\rho = -0.00, p = 1.00, N = 50$ | |
| Ms. Pacman | $\rho = -0.26, p = 0.56, N = 50$ | |
| Minipacman | $\rho = -0.06, p = 1.00, N = 50$ | |
| Sokoban | $\rho = -0.02, p = 1.00, N = 50$ | |
| 9x9 Go | $\rho = 0.52, p = 0.06, N = 25$ | . |

Table 15: Effect of the training search budget, $B$, on the strength of the policy prior, modeled as `Reward ~ Environment * log(B) + log(B)`$^2$ over $N = 450$ data points. Top: this ANOVA indicates that the environment and budget are significant predictors of reward, and that there is a second-order effect of the search budget, indicating that performance goes down with too many simulations. Additionally, there is an interaction between environment and budget. Bottom: individual Spearman rank correlations between reward and $\log(B)$ for each environment. $p$-values are adjusted for multiple comparisons using the Bonferroni correction. Note that the correlation for Go does not include values for $B > 50$ (and thus is largely flat, since Go does not learn for small values of $B$).

| Variable | Statistic | Strength of Evidence |
|---|---|---|
| Environment | $F(7, 433) = 45.45, p < 0.001$ | *** |
| $\log(B)$ | $F(1, 433) = 593.90, p < 0.001$ | *** |
| $\log(B)^2$ | $F(1, 433) = 121.70, p < 0.001$ | *** |
| Environment:$\log(B)$ | $F(7, 433) = 11.64, p < 0.001$ | *** |
| Acrobot | $\rho = 0.76, p < 0.001, N = 60$ | *** |
| Cheetah | $\rho = 0.76, p < 0.001, N = 60$ | *** |
| Humanoid | $\rho = 0.88, p < 0.001, N = 60$ | *** |
| Hero | $\rho = 0.75, p < 0.001, N = 60$ | *** |
| Ms. Pacman | $\rho = 0.61, p < 0.001, N = 60$ | *** |
| Minipacman | $\rho = 0.70, p < 0.001, N = 60$ | *** |
| Sokoban | $\rho = 0.87, p < 0.001, N = 60$ | *** |
| 9x9 Go | $\rho = 0.48, p = 0.05, N = 30$ | . |

## D.7 PLANNING FOR GENERALIZATION

Table 16: Effect the evaluation search budget, $B$, on generalization reward when using the learned model with MCTS, modeled as `Reward ~ Environment * log(B)` over $N = 350$ data points. Top: this ANOVA indicates that the environment and budget are significant predictors of reward, and that there is an interaction between environment and budget. Bottom: individual Spearman rank correlations between reward and $\log(B)$ for each environment. $p$-values are adjusted for multiple comparisons using the Bonferroni correction.

| Variable | Statistic | Strength of Evidence |
|---|---|---|
| Environment | $F(6, 336) = 13.03, p < 0.001$ | *** |
| $\log(B)$ | $F(1, 336) = 9.31, p = 0.002$ | ** |
| Environment:$\log(B)$ | $F(6, 336) = 31.50, p < 0.001$ | *** |
| Acrobot | $\rho = 0.77, p < 0.001, N = 50$ | *** |
| Cheetah | $\rho = 0.25, p = 0.58, N = 50$ | |
| Humanoid | $\rho = 0.32, p = 0.17, N = 50$ | |
| Hero | $\rho = -0.12, p = 1.00, N = 50$ | |
| Ms. Pacman | $\rho = -0.23, p = 0.73, N = 50$ | |
| Minipacman | $\rho = -0.39, p = 0.03, N = 50$ | * |
| Sokoban | $\rho = 0.72, p < 0.001, N = 50$ | *** |

Table 17: Effect the evaluation search budget, $B$, on generalization reward when using the simulator with MCTS, modeled as `Reward ~ Environment * log(B)` over $N = 350$ data points. Top: this ANOVA indicates that the environment and budget are significant predictors of reward, and that there is an interaction between environment and budget. Bottom: individual Spearman rank correlations between reward and $\log(B)$ for each environment. $p$-values are adjusted for multiple comparisons using the Bonferroni correction.

| Variable | Statistic | Strength of Evidence |
|---|---|---|
| Environment | $F(6, 336) = 87.02, p < 0.001$ | *** |
| $\log(B)$ | $F(1, 336) = 271.62, p < 0.001$ | *** |
| Environment:$\log(B)$ | $F(6, 336) = 89.38, p < 0.001$ | *** |
| Acrobot | $\rho = 0.82, p < 0.001, N = 50$ | *** |
| Cheetah | $\rho = 0.76, p < 0.001, N = 50$ | *** |
| Humanoid | $\rho = 0.48, p = 0.003, N = 50$ | ** |
| Hero | $\rho = 0.08, p = 1.00, N = 50$ | |
| Ms. Pacman | $\rho = 0.97, p < 0.001, N = 50$ | *** |
| Minipacman | $\rho = 0.46, p = 0.005, N = 50$ | ** |
| Sokoban | $\rho = 0.95, p < 0.001, N = 50$ | *** |

Table 18: Rank correlations between the search budget, $B$, and generalization reward in Minipacman for different types of mazes and models. $p$-values are adjusted for multiple comparisons using the Bonferroni correction.

| Scene Type | Model Type | Correlation | Strength of Evidence |
|---|---|---|---|
| In-distribution | Learned model | $\rho = -0.54, p < 0.001, N = 180$ | *** |
| Out-of-distribution | Learned model | $\rho = -0.54, p < 0.001, N = 180$ | *** |
| In-distribution | Simulator | $\rho = 0.25, p = 0.002, N = 180$ | ** |
| Out-of-distribution | Simulator | $\rho = 0.33, p < 0.001, N = 180$ | *** |

Table 19: Effect the evaluation search budget ($B$), the number of unique training mazes ($M$), and test level on generalization reward in Minipacman when using the simulator with MCTS, modeled as `Reward ~ log(M) * log(B) + Test Level` over $N = 360$ data points. This ANOVA indicates that the both the number of training mazes and the search budget are significant predictors of reward, and that there is an interaction between them.

| Variable | Statistic | Strength of Evidence |
|---|---|---|
| Test Level | $F(1, 355) = 11.23, p < 0.001$ | *** |
| $\log(M)$ | $F(1, 355) = 660.98, p < 0.001$ | *** |
| $\log(B)$ | $F(1, 355) = 204.29, p < 0.001$ | *** |
| $\log(B){:}\log(M)$ | $F(1, 355) = 108.80, p < 0.001$ | *** |

