# OpenReview forum: "On the role of planning in model-based deep reinforcement learning"
_ICLR.cc/2021/Conference — ICLR 2021 Poster_

### Official Review · AnonReviewer3 · 2020-10-20
**Good ablation study that provides some insights on the role of planning in MBRL.**

**Rating:** 7
**Confidence:** 3

**Review:**

Summary:

This paper tries to disentangle the role of planning in model-based reinforcement learning with a number of different ablations and modifications to MuZero. Specifically, the authors analyze the overall contribution of planning by omitting planning from which it is originally used in MuZero, and investigate different planner settings that can drive performance. In addition, they check the generalization advantage of MBRL. Overall, the paper is well-written, and experiments are conducted appropriately. The results provide some insights that other researchers in the MBRL community can leverage for their future work. My major concern is the lack of direct ablation study that can clearly show the advantage of planning in providing a good learning signal. See the detailed comments below.

Comments:

* To bolster the argument that planning contributes the most in providing a good learning signal, “(Data)” or “(Data+Eval)” ablation can be done:  training update is done with targets generated with MCTS of $D_{\text{UCT}} = D_{\text{tree}} = 1$ while the agent uses full MCTS during training rollouts. Since the current experiments do not ablate planning in training, the effect of planning is only implicitly examined.
* In the extreme, model-free version of “(Data)” could be implemented in which $q$ is trained instead of $v$ while $v^{\text{MCTS}}$ and $\pi^{\text{MCTS}}$ is replaced with $\max q$ and $\arg\max q$. If this change is viable, it would also provide a chance to confirm the effect of planning for exploration only.
* BFS is a too weak planner to compare with. How about random shooting methods?

Styles & Typo:

* Use “\citet” when appropriate; when a citation is used as a subject in the sentence.
* Line 3 of Algorithm 2 in Appendix: $k=1 \dots S$ → $k=1 \dots B$ ?

---

> ### Author Response · Authors · 2020-11-17
> **We are running experiments with your suggested "Data" ablation**
>
> Thank you for your thoughtful review; we are glad that you liked the paper and that you found it insightful!
>
> 1. That’s a great suggestion to perform updates with $D_\mathrm{tree}=D_\mathrm{UCT}=1$ while acting according to the full search. We have implemented this and are in the process of running experiments. Please also see our longer discussion of this in the response to all reviewers (see “Contributions of planning in computing policy updates”).
>
> 2. The model-free version of “(Data)” implemented via Q-functions is also an interesting idea, and is quite similar to what was explored by Hamrick et al. (2020) “Combining Q-learning and search with amortized value estimates”. In their work, they found that this version was actually quite unstable (see Figure 3, “SAVE w/o AL”). We will add some discussion of this to the text.
>
> 3. We chose to compare to BFS not because of its strength or weakness as a planner but to understand how the model and value function behave when they are used differently than during training (though we note that with large amounts of search, BFS is the strongest possible planner, as it amounts to exhaustive search). We actually find the BFS results to be some of the most interesting results in the paper, because they highlight the substantial mismatch between the actions recommended by the value function and the policy prior. For example, in Minipacman and Sokoban, the branching factor is 5; therefore 5 steps of BFS search means that all actions at the root node are expanded and the action will be selected as the one with the highest value. Yet, the performance of this agent is substantially worse than just relying on the policy prior. For sparse reward environments, any planner that relies on the value function without the policy prior will suffer from this effect, including random shooting. We will clarify the significance of this comparison in the text.
>
> Thank you for catching the typos and style issues, we will fix these in the text as well.

---

### Official Review · AnonReviewer2 · 2020-10-28
**Needs to rephrase claims and design experiments more carefully.**

**Rating:** 5
**Confidence:** 4

**Review:**

Overview of the paper:
This paper studies three empirically questions about the planning part in Muzero, an algorithm integrating direct learning, model-learning, and planning. These three questions, as written in the paper, are that 1) for what purposes is planning most useful? 2) what design choices in the search procedure contribute most to the learning process 3) does planning assist in generalization across variations of the environment. The paper answers all these three questions using experiments: 1) the major reason for the performance improvement using planning is maintaining a policy that approximates the policy found by the MCTS algorithm, 2) simpler and shallower planning is often as performant as more complex planning, 3)  search at evaluation time only slightly improves zero-shot generalization.

Comments:
Overall I think the paper is not ready to publish.

While the 3 questions asked in the paper are quite general and apply to many model-based algorithms, the paper provided general answers to them using empirical results only for the Muzero algorithm under deterministic environments, which doesn't look appropriate to me. The Muzero algorithm is different from many other planning algorithms, such as Dyna-style planning, MPC, value iteration, etc. And deterministic environments are easy cases compared with stochastic or even partial observable environments. I would suggest the authors rephrase the questions and answers to make them more specific so that the results in the paper can support conclusions well.

While there are multiple things making me confused, I would like to highlight the following one as an example because that almost makes me think their answer to their first question is incorrect. The answer is drawn from the results shown in figure 3, which illustrates how important three design choices (following MCTS policy in both training and testing, following MCTS policy in training and prior policy in testing, and following prior policy in both training and testing) are in planning. These results could tell us how much more performance the algorithm achieves by following MCTS policy in training or testing, but they can not tell us how much more performance the algorithm achieves by updating the prior policy towards the MCTS policy compared with other approaches. That is, in all cases, the algorithm updates its prior policy towards the MCTS policy.

The other thing I feel not appropriate is, while the paper claimed that "we systematically study the role of planning and its algorithmic design choices in a recent state-of-the-art MBRL algorithm, MuZero", the planning part is not the only part that varies across different design choices. In particular, because the model parameters are learnable and all the variations of algorithms they tested have different updates to model parameters. The resulting learned models are different in these variations. Thus it is not appropriate to conclude that the performance difference between different variations is solely the result of planning. It might also come from differences in learned models.

Thanks for the author's detailed response.

In terms of the first question, I do appreciate the value of the paper as a nice empirical study of Muzero and other similar MBRL algorithms. Meanwhile, many MBRL algorithms are not like Muzero. For example, value-based planning algorithms don't maintain an explicitly parameterized policy. Therefore the conclusions here may not apply to all cases. Making its conclusions more precisely will not undermine the value of the paper, instead, it provides readers clearer results. I do see in the revised version, the authors changed their language in the discussion about the result. But maybe clearer results themselves are better.

My second concern is addressed in the updated version of the paper, with additional experiments. Cool!

In terms of the third question, after reading your response, I think there is a very interesting question. When we test planning algorithms, should we give the agent a fixed model and a fixed representation or fixed algorithms learning the model and the representation? After thinking for a while, I can see the advantages and disadvantages of both cases. So I would change my mind and agree with the authors that their choice of testing is valid. But I do hope this choice being mentioned in the paper because people like me would typically consider the other one.

I would like to raise my score to 5.

---

> ### Author Response · Authors · 2020-11-17
> **Our work provides important insights for integrated MBRL agents**
>
> Thank you for your time in reviewing our paper. We hope that we can convince you of the merits of our work; we believe there may be some misunderstandings in what our paper aims to do, which we attempt to address here. We also note that the other three reviewers all found our results to be insightful and thorough, which we think is an indicator of their potential impact in the wider community.
>
> We would like to begin by clarifying that the point of our paper is not to explain why MuZero is better than other algorithms, but to understand how MuZero’s performance is affected when using planning in different ways and with different design choices. This type of analysis is important because while MuZero is state-of-the-art in MBRL in terms of final reward obtained, it is a very complex algorithm and the roles and interactions of its different components are not well understood. Our experiments allow us to better understand the role of planning in its design, which we think is important both for understanding MuZero itself and building intuition about the role of planning in MBRL more generally. We believe that our conclusions---for example, that standard environments may not be really measuring the capacity of agents to “reason”---are important takeaways that should inform future research in MBRL.
>
> Responding to your specific points:
>
> 1. Regarding the generality of MuZero and the types of environments studied, please see our response to all reviewers. MuZero has important connections to all the algorithms you listed (and in fact uses MPC as a subroutine), and the environments on which we evaluated are also very standard in the MBRL literature. However, we acknowledge that we haven’t directly tested all possibilities, and are therefore working to adjust the language in the paper to be more precise regarding our claims.
>
> 2. While it is true that all our experiments in Figure 3 use planning to update the policy, we disagree that this invalidates our claims. However, we do agree our claims could be strengthened by the addition of further baselines as suggested by R3, which we are working on implementing. Please see our response to all reviewers (“Contributions of planning in computing policy updates”).
>
> 3. MuZero is a complex, integrated agent with many working parts, and like many other MBRL algorithms (MBPO, Dreamer, etc.), the model is learned online and therefore affected by any changes which affect learning. As we are interested in understanding the role of planning on the behavior of the whole integrated MBRL system (i.e., planning, learning, and the interactions between them), we believe it is appropriate to experiment on the planning procedure while keeping the rest of the agent untouched. It is not a problem if modifications to planning affect model learning, because the model is part of the whole system and we are interested in the whole system’s behavior. Consider that even some model-free algorithms have this property: for example, in actor-critic agents, changes to the behavior policy will affect learning of the critic which will affect performance. But this property does not invalidate hypotheses or claims about, for example, the effect of exploration on the overall agent’s performance. Similarly, just because changes to the planning procedure result in different models does not mean that we cannot draw conclusions about the effect of those changes on overall behavior.
>
> Your comment makes it clear, however, that we were not clear enough in the paper that we are interested in the role of planning within integrated MBRL agents. We will clarify this and add some discussion about how the various components in agents like MuZero interact.
>
> We also think that your point about the effect of the model quality on agent performance suggests two very interesting follow-up questions, which we hope that future work will address by building on our paper! First, it would be interesting to perform a separate analysis of how changes to the planning procedure affect the quality of the learned model. Second, it would also be interesting to understand the contributions of planning when given a pretrained, fixed model (though it’s not immediately clear the best way to do this in MuZero, given that the policy, value, and dynamics share a common torso). By using a pretrained model, this would allow a deeper understanding of how the model quality influences planning performance. (We do note that other recent work has already looked at this question to some extent (e.g. Janner et al., 2019), and some of our generalization experiments touch on it as well (Figure 5b, 5c, 6)).
>
> We are happy to clarify any remaining questions you may have!

---

### Official Review · AnonReviewer1 · 2020-10-28
**Good analysis for the role of search in MBRL**

**Rating:** 6
**Confidence:** 4

**Review:**

summary:
This paper analyzes the role of planning in the model-based reinforcement learning agent, based on evaluating MuZero on eight tasks (i.e. Ms.Pacman, Hero, Minipacman, Sokoban, 9x9Go, Acrobot, Cheetah, and Humanoid), which have discrete action spaces. The conducted experiments show three major implications: (1) Of the three parts in which search is used (i.e. search at evaluation time, search at training time for exploration, and using search result as a policy target), the role of serving as a policy improvement target was most substantial. (2) Deep tree search did not make a significant contribution to performance, and a simple Monte-Carlo rollout could be performant enough for MBRL. Also, a too small or too large search budget can be harmful to the performance of the MBRL agent. (3) Search at evaluation time was helpful for zero-shot generalization especially when the model is accurate.


pros:
- The paper is well-written and well-organized. Hypotheses and the experiments are well-designed and seem thorough.

cons:
- The analyses are limited to MuZero that deals with discrete action space and to the deterministic environments. Since MuZero is a particular instance of MBRL, where only the reward and value prediction are performed in the latent-state space, it is unclear that the conclusions of the work can be applied to other instances of MBRL (e.g. MBRL methods dealing with dynamics model that operates on the original state space). I am not fully convinced that the results here can be generalized to other classes of MBRL.


some questions and comments:
- It is interesting that the main benefit of the search is serving as a policy target. What are the advantages of policy improvement through search in MuZero compared to directly optimizing policy using analytic gradients in latent space like Dreamer (Hafner et al, 2020)?

- In Figure 4c, why does the performance get worse as the number of simulations increases in Acrobot and Cheetah? Or in Figure 6 (Simulator, Green line), why the performance of (# Simulations = 3125) is worse than (# Simulations = 125)? This may not be explained as compounding model errors since the agent is using the exact simulator.

- It would be have been great to see the learning curve even for the results of Section 4.3, similarly to Appendix D2 that presents the learning curve for the results of Section 4.2.

---

> ### Author Response · Authors · 2020-11-17
> **We are running more experiments with pixel reconstruction loss**
>
> Thank you for your review! We are glad you find the paper to be well-written and the experiments to be well-designed and thorough.
>
> With regards to your main question about applicability to MBRL more generally, please see our response to all reviewers.
>
> It’s a great question how our results might change when learning dynamics over the original state space! In the course of developing our experiments for this paper, we actually performed a preliminary test of this question (by adding a pixel reconstruction loss to MuZero) but did not find that reconstructing observations made much of a difference. We are working on more formally re-running some of our experiments in Sokoban using the pixel reconstruction loss and will include the results in the Appendix.
>
> Regarding the comparison of search versus gradient-based planning (i.e. in SVG-style methods like Dreamer), we agree that it would be very interesting to test this. We believe that this question highlights one of the strengths of our work, which is that it brings into focus a number of follow-up questions about how planning can best serve MBRL agents such as “what types of planning lead to the most reliable policy and value targets?”. Additionally, please see our discussion of some further connections to SVG in our response to all reviewers (see “Generality of MuZero”).
>
> In Figure 4c, the performance of some agents decreases with large search budgets due to compounding model errors. This is an instance of the same problem discussed by a number of other papers, such as MBPO (Janner et al., 2019), and can be also seen in Figure 3b of the MuZero paper (Schrittwieser et al, 2019). The models learned for different environments will be of varying qualities, which is why we do not see this effect in all environments. If we were to continue increasing the search budget, we would likely eventually see a drop-off everywhere.
>
> In Figure 6, the performance decreases even with the simulator due to off-policy errors in the value function, which get worse with larger searches. In fact, this is the same effect as what can be seen in Figure 5c; the only reason it is less drastic than with BFS is because the policy prior used by MCTS keeps the search more on-policy. Note that Figure 5b does not demonstrate this effect because the training and testing distributions are the same (though we expect with large enough search budgets, we would see decreases here too); in contrast, in Figure 6, the errors in the value function are more severe because the agent is being asked to generalize to out-of-distribution scenes. We will clarify these points in the text.
>
> We will update the Appendix to include learning curves for the results in Section 4.3.

---

### Official Review · AnonReviewer4 · 2020-10-28
**Official Blind Review #4**

**Rating:** 7
**Confidence:** 4

**Review:**

The paper investigates how and why planning might be beneficial in model-based reinforcement learning settings. To that end, the authors ask three questions on planning in MBRL: (1) How does planning benefit MBRL agents? (2) Within planning, what choices drive performance? (3) To what extent does planning improve generalization? In order to answer these questions, the authors investigate the performance of MuZero in a variety of learning challenges while systematically ablating the algorithm to find how each part of the algorithm effects the overall performance.

In its current form the paper is marginally below the acceptance threshold.

The reasoning for my judgement is as follows:
The paper makes strong claims about how planning effects model-based RL algorithms based on their experimental results. However, the results are only based on five different seeds. Given the fundamental conclusions the authors formulate, the results are not statistically relevant enough. (For further intuition about the impact of a small number of seeds in DRL, please refer to ‘Deep Reinforcement Learning that Matters’, Figure 5 by Henderson 2017). The strong language in combination with the limited statistical relevance result in the experimental design not being sufficient.

Having that said, the paper itself is well written and provides important insights to the community. I would strongly encourage the authors to either adjust the language or (better) run additional experiments to strengthen the paper. With more runs for each of the experiments, I would recommend a clear accept.

Minor comment: At the beginning of the second paragraph, the authors state ‘Many have suggested that models will play a key role in generally intelligent artificial agents’. The supporting papers are essentially just two different author-sets, making this another instance of too strong of a statement with too little foundation.

-----------------------------------------------------------------------------------------
After reading the authors response to all reviewers, I believe all questions to be sufficiently addressed. I will therefore (happily) raise my score.

---

> ### Author Response · Authors · 2020-11-17
> **Running 5 more seeds of each experiment**
>
> Thank you for your review. We are very glad you found our paper to be clear and its insights to be important!
>
> As discussed in our response to all reviewers (see “Strength of claims”), we are working on adjusting the language in the text to be more precise regarding the claims being made.
>
> We agree that statistical power is an important requirement for making empirical claims, and that using 5 seeds to perform a t-test is insufficient, as in the example from Henderson et al. (2017). However, we note that our conclusions are based mostly on Spearman rank correlations which pool across multiple parameter settings or on comparisons that pool across multiple environments. For example, the result that the “Learn” variant of MuZero achieves 69.4% of vanilla MuZero is computed from 40 datapoints, and the individual rank correlations computed for each environment (e.g. for D_tree, D_UCT) use 25 datapoints. Please see Tables 11-18 in the Appendix for statistical tests, which we will also update to include sample sizes. Overall, we trained 660 unique agents and ran a further 1060 test-time experiments, which we believe is already quite extensive. Of course, it can never hurt to increase statistical power and as you have pointed out, doing so will strengthen our results. Thus, we are running 5 more seeds for each experiment (for a total of 10). We do not anticipate the results will change our overall findings, however, for the reasons described above.
>
> We will change “Many” to “Some” in the second paragraph, and will also add a few more references from other authors.

---

> > ### Comment · AnonReviewer4 · 2020-11-17
> > **Good author response**
> >
> > Dear Authors,
> >
> > Thank you for the response in general and the response to my questions in particular. I appreciate the further explanation and the hint towards the tables 11-18 as well as more runs. Additionally, I believe the answers to the other reviewers to be sufficient as well.
> >
> > From my point of view, all questions have been addressed and I am happy to raise my score.

---

### Author Response · Authors · 2020-11-17
**General response to reviewers (1/2)**

Dear reviewers, thank you for your time in assessing our paper and for your feedback, which has helped us to improve the paper. We hope we are able to address your concerns and also clear up some misconceptions. As there were a few concerns shared by multiple reviewers, we address those here, and limit direct replies to more specific comments.

We would first like to emphasize the scale, breadth, and potential impacts of our study. We report results for 1720 experiments (660 agents trained and a further 1060 test-time experiments) in which we train the highly complex MuZero architecture. Our experiments span eight unique environments in five distinct domains that have very different characteristics. Rather than simply testing different hyperparameter settings, we formulated specific, targeted ablations to test our hypotheses in the first large-scale study analyzing MuZero’s performance. Our results have important implications regarding the complexity of planning required to achieve good performance, the degree to which standard environments measure reasoning abilities, and the potential for planning to help with generalization. Finally, several reviewers raised interesting questions after reading our paper, which we see as being an indication that our work generates important insights and will stimulate follow-up research.

*Summary of changes*

We are working on implementing a number of changes to improve the paper, summarized below, and will upload a revised version of the paper later this week.

We are running the following additional experiments as suggested by various reviewers. While we will unfortunately not be able to have all the results by the end of the discussion period due to resource limitations, we will update the paper with results as they complete:

- Running 5 more seeds of every experiment (for a total of 10), to increase statistical power
- Running a version of MuZero with pixel reconstruction loss, to test the effect of model learning on the original state space
- Running a “Data”-only version of MuZero with $D_\mathrm{tree}=D_\mathrm{UCT}=1$ for learning and either full or no MCTS for acting, to further explore the role of search on learning and on the data distribution

We will also make the following modifications to the text:

- Addition of supplementary figures showing learning curves for Section 4.3
- Inclusion of sample sizes used for all statistical tests
- Clarify the language around the generality and scope of our results
- Expand on the relationship between MuZero and other MBRL methods
- Miscellaneous smaller clarifications and typos

*Determinism and partial observability*

R1 and R2 also felt that our emphasis on (mostly) deterministic environments limited the paper. However, we would like to clarify that the domains we tested are not all fully deterministic or observable: the movement of the ghosts in Minipacman is stochastic, 9x9 Go is a two-player game and therefore neither deterministic nor fully-observed from the point of view of each player, and the limited number of frames in Atari makes it partially observed (in particular, events like the ghosts in Ms. Pacman changing from edible to dangerous cannot be predicted). We will clarify this in the main text.

Moreover, the environments we have focused on are very standard in the literature and the majority of work in MBRL (and much of RL, more generally) focuses on Mujoco and/or Atari (e.g., MuZero, Dreamer, and MBPO). Our results therefore are relevant to the work being done by other researchers in the community, even if the focus is on (mostly) deterministic environments.

---

### Author Response · Authors · 2020-11-17
**General response to reviewers (2/2)**

*Generality of MuZero*

R1 and R2 both questioned whether our results with MuZero can be generalized to other MBRL methods. Although our results of course do not provide direct evidence about how other MBRL methods behave, we believe that our experiments can provide good intuitions and serve as a blueprint for future investigations. We emphasize that there are many connections between MuZero and other methods in the Dyna, MPC, Value/Policy Iteration, and SVG families, as described in the next paragraph. Finally, given that MuZero is the state of the art MBRL algorithm in the majority of the domains we tested (in terms of final reward achieved), we believe that it is important to understand the details of what drives its performance so that these insights can be distilled.

- Dyna: Many recent MBRL methods rely on a form of Dyna-style planning that simulates data from the model and then updates the policy on this data using TRPO (including MBPO, ME-TRPO, SLBO, and PAL).  While this may sound superficially different than the update MuZero uses, recent work by Grill et al. (2020) showed that the MuZero policy update approximates TRPO. Thus, our “Learn” variant of MuZero (Section 4.2) is very similar to these other Dyna-style methods. The primary difference is that MuZero performs multiple rollouts from the same state, while Dyna-style methods typically perform only one. This difference is interesting and we hope that future research will explore its implications.
- MPC: MuZero uses MPC for acting, and many of our experiments explicitly focus on the behavior of MuZero in this regime (i.e., at evaluation time only). Indeed, the results in Section 4.4 and Figures 5 and 6 could alternately be titled “Contributions of MPC”.
- Value and Policy Iteration: The vanilla forms of value and policy iteration are not tractable in the domains we study, due to their requirement of sweeping over the entire state space. However, approximate forms of these methods can prove useful. Indeed, MuZero with MCTS implements approximate K-step policy iteration (Efroni et al., 2019), and our BFS implementation is equivalent to local, depth-limited value iteration.
- Value gradients: The use of value gradients (as in Fairbank et. al, 2008, Heess et al. 2015, Byravan et. al, 2020) leverages model gradients to estimate a policy gradient estimate; this estimate is similar (without regularization) to the one MCTS as a policy iteration operator converges to (Grill et al., 2020). There are of course some differences: for example, model gradient methods can only be used in an unbiased fashion for continuous actions domains, and several works (see e.g. Balduzzi et. al, 2015 and D’Oro et. al, 2020) point out that the loss typically used to estimate models cannot guarantee the model gradients themselves are trustworthy. While we agree it would be interesting to more extensively test the impacts of these differences in practice, we believe this question is better suited to future work.

We will add the above discussion to the main text as well to clarify this point.

*Contributions of planning in computing policy updates*

R2 and R3 questioned whether we have sufficiently shown that planning helps the most by providing targets for policy and value learning. Indeed, we did not show any results during training where planning is used for acting but not learning. This was largely due to the fact that it is not clear to us what should be used for learning instead; different choices of update rule (e.g. TRPO, MPO, A2C, etc.) can have drastically different effects which would make it hard to isolate the role of planning in acting on top of these. Moreover, our aim was not to compare to model-free approaches, but to characterize the different ways in which planning can be used within the agent.

R3 suggested an interesting ablation, however, which is to update the policy using the $D_\mathrm{tree}=D_\mathrm{UCT}=1$ variant of MCTS while acting according to full MCTS. Because $D_\mathrm{tree}=D_\mathrm{UCT}=1$ is very close to model-free (it can be implemented with a Q-function), this brings us closer to a variant which (mostly) does not use search for learning. We have implemented this variant and are in the process of running new experiments on all environments. To isolate the effect of the shallow update on its own, we will also include a baseline which uses $D_\mathrm{tree}=D_\mathrm{UCT}=1$ during learning while acting from the policy prior.

*Strength of claims*

R1, R2, and R4 all felt that the language used to present our claims was too strong. While we do believe our results have implications for MBRL more generally, we take the point that they do not provide direct evidence. We will adjust the language in the paper to be clearer about this.

---

### Author Response · Authors · 2020-11-24
**New revision uploaded**

Dear reviewers, thank you again for your time in assessing our paper and your constructive feedback. Based on this, we have run several additional experiments which we feel have improved the strength of the paper’s claims, providing more nuanced and precise results. We have uploaded a revised version of the paper with these new results, which we also summarize here (though we note that a few experiments are still running and are thus excluded from the revised paper). Specifically:

- We have run 5 extra seeds for every experiment, which have not changed any of our conclusions.

- We have included some experiments in the appendix (Section D.2) where we train MuZero’s model to reconstruct observations via an additional reconstruction term in the loss. Overall, our results indicate that learning a model in observation space does not fundamentally alter the role of planning in MuZero, though it does improve the fidelity of the model (which can benefit planning). This result helps to strengthen our claim that our overall findings are useful in building intuition about other related MBRL methods which use models learned in the observation space.

- We have included two new ablations in Section 4.2: “One-Step” (which uses $D_\mathrm{tree}=1$ for policy updates, and acts according to the policy prior) and “Data” (which uses $D_\mathrm{tree}=1$ for policy updates, and acts according to $D_\mathrm{tree}=\infty$). The “One-Step” variant allows us to establish the performance of MuZero when using the model as minimally as possible (in principle, it could be implemented without a model at all and just with a Q-function). This version still obtains reasonable performance in many tasks, replicating our results with $D_\mathrm{tree}=1$ from Section 4.3, though still underperforms “Learn”. The “Data” variant allows us to identify that planning plays complementary roles for computing policy updates and for generating a useful data distribution, and that “Learn+Data” has such strong performance (~90% of full MuZero) due to it combining both of these strengths.

We have also improved the writing of the paper with the following additional changes:

- Addition of supplementary figures showing learning curves for Section 4.3
- Inclusion of sample sizes used for all statistical tests
- Clarify the language around the generality and scope of our results
- Expand on the relationship between MuZero and other MBRL methods
- Miscellaneous smaller clarifications and typos

---

### Decision · Program_Chairs · 2021-01-07
**Final Decision**

**Decision:**

Accept (Poster)

**Comment:**

The reviewers appreciated the author replies, the additional experiments (more runs but also more ablations/baselines), and the updated paper. Also R2 is now largely satisfied (but seems to have been too late to post a public reply or to raise the score of the review).

The paper provides important insights in model-based RL and its connections to planning, by studying MuZero with systematic ablations. Hence a valuable contribution to the community. All (major) cons have been addressed in the revision.